# Updating Inventory, Deformation, and Development Characteristics of Landslides in Hunza Valley, NW Karakoram, Pakistan by SBAS-InSAR

Xiaojun Su [1,2,3], Yi Zhang [1,2,3,*], Xingmin Meng [1,2,3], Mohib Ur Rehman [1,2,3,4], Zainab Khalid [1,2,3,5] and Dongxia Yue [1,2,3]

1 MOE Key Laboratory of Western China's Environmental Systems, School of Earth Sciences, Lanzhou University, Lanzhou 730000, China
2 Gansu Technology Innovation Centre for Environmental Geology and Geohazard Prevention, School of Earth Sciences, Lanzhou University, Lanzhou 730000, China
3 College of Earth and Environmental Sciences, Lanzhou University, Lanzhou 730000, China
4 Department of Earth Science, COMSATS University Islamabad, Abbottabad Campus, Abbottabad 22010, Pakistan
5 Department of Development Studies, COMSATS University Islamabad, Abbottabad Campus, Abbottabad 22010, Pakistan
* Correspondence: zhangyigeo@lzu.edu.cn

**Abstract:** The Hunza Valley, in the northwestern Karakoram Mountains, North Pakistan, is a typical region with many towns and villages, and a dense population and is prone to landslides. The present study completed landslide identification, updating a comprehensive landslide inventory and analysis. First, the ground surface deformation was detected in the Hunza Valley by SBAS-InSAR from ascending and descending datasets, respectively. Then, the locations and boundaries were interpreted and delineated, and a comprehensive inventory of 118 landslides, including the 53 most recent InSAR identified active landslides and 65 landslides cited from the literature, was completed. This study firstly named all 118 landslides, considering the demand for globally intensive research and hazard mitigation. Finally, the deformation, spatial–topographic development, and distribution characteristics in the Hunza Valley scale and three large significant landslides were analyzed. Information on 72 reported landslides was used to construct an empirical power law relationship linking landslide area ($A_L$) to volume ($V_L$) ($V_L = 0.067 \times A_L^{1.351}$), and this formula predicted the volume of 118 landslides in this study. We discovered that the landslides from the literature, which were interpreted from optical images, had lower levels of velocity, area, elevation, and height. The SBAS-InSAR-detected active landslide was characterized by higher velocity, larger area, higher elevation, larger slope gradient, larger NDVI (normalized difference vegetation index), and greater height. The melting glacier water and rainfall infiltration from cracks on the landslide's upper part may promote the action of a push from gravity on the upper part. Simultaneously, the coupling of actions from river erosion and active tectonics could have an impact on the stability of the slope toe. The up-to-date comprehensive identification and understanding of the characteristics and mechanism of landslide development in this study provide a reference for the next step in landslide disaster prevention and risk assessment.

**Keywords:** landslide; InSAR; landslide inventory; slope displacement; Hunza; Pakistan

## 1. Introduction

Landslides are one of the most common and widespread natural hazards, and they cause major damage to properties, infrastructures, and communities, as well as numerous casualties worldwide [1–3]. The Indus River Basin in northern Pakistan hosts high mountains and is prone to multiple hazards [4,5]. In this region, the Hunza Valley with a high

population density is one part of Gilgit-Baltistan and has been marked by its vulnerability to multiple geo-hazards including landslides, avalanches, and flash floods. On 4 January 2010, a huge landslide triggered by thrust fault activities caused 20 fatalities at Atta Abad and formed a 120 m high dam [6–8]. The landslide deposition blocked the Hunza River and subsequently impounded a reservoir with a length of 21 km, and the Karakoram Highway along the bank of Hunza River was disrupted [8]. Recently, a catastrophic rock-ice avalanche on 7 February 2021 at Chamoli in the Indian Himalayas caused severe damage to the ecology and safety of downstream areas [9]. A rock avalanche occurred in the channel of the Boultar Glacier within the Hunza River watershed in 1986, which was the first rock avalanche event reported in Karakoram [10]. The deposits of the rock avalanche covered an area of 3.5 km$^2$ and were transferred by glaciers for many years. The Ganesh-Saukien is another example of a rock avalanche [8,11]. Widespread landslides expose this area to the potential for building collapse and ever-present damage. These catastrophes usually occurred without warning and were not predicted by researchers, resulting in underestimated adverse consequences and loss. A comprehensive landslide inventory, however, would provide the fundamental information base for further assessment of landslide susceptibility and risk [12]. Knowledge of the location and distribution of landslides is vital for predicting affected areas and for quantifying hazard risk. An up-to-date landslide inventory of the Hunza Valley including the deformation velocity, pattern of the deformation process, and field surveyed features would be significant.

Landslide identification and a survey on a regional scale have been the main challenges for geologists and geological hazards managers in recent decades. The technology used for landslide surveys has experienced a long-term evolution and advancement with the development of surveying and mapping science and satellite technology [13–15]. In past decades, because the technology was backward, landslide investigation was conducted with limited available technologies. Unlike the previous era, researchers can now utilize the optimal instruments and techniques to design the research and missions. Various techniques including GNSS, remote sensing, satellite radar, and antenna or ground-based SAR have been used in landslide identification [16], routine monitoring [17], early warning [14], and risk emergency response [18].

To some extent, many disasters occur because of the lack of coordination between management measures and the mapping and zonation of hazards and risk [19,20]. It has been reported that errors in the input data of landslide inventory maps are a major limitation on the reliability of landslide hazard assessment [21]. In and around the study area, the effects of landslide inventories prepared from satellite data on landslide hazard assessment in the High Himalayas terrain of India have been discussed [22]. Landslide inventory mapping in the Hunza Valley region was conducted based on image interpretations or direct field surveys without referring to recent deformation information [6,8]. The goal of this study is to acquire the location; velocity of recent ground surface deformation, updated landslide inventory; and the deformation, development, and distribution of landslides in the Hunza Valley using multi-look SAR datasets.

SBAS-InSAR (Small Baseline Subsets-Synthetic Aperture Radar Interferometry) is an advanced remote sensing technique that has been widely applied in displacement monitoring and the detection of geo-hazards in mountainous regions [23–25]. Displacement rates during the monitoring period are potentially capable of analyzing the landslide failure process on a single slope unit. A comprehensive landslide inventory produced using the combination of time-series InSAR, visual interpretation, and field investigations can potentially determine the susceptibility of terrain to the occurrence and development of landslides [26]. Similarly, there is a demand to determine the slope risk on a regional scale using InSAR monitoring data. Landslide, in general, is a geological process that causes a slow-moving deformation over a long period or a rapid, large deformation in a relatively short time. It eventually experiences failure in the types of slide, flow, fall, and topple along the slip surface or structures such as a soft layer or joint plane under various causes such as gravity, earthquakes, rainfall, and human activity or a combination of these factors [27].

The slope consequently slides integrally or segmentally. In this way, the slope deformation in the long term is a key point in the monitoring and intensive investigation. Sentinel-1 SAR data can support operational applications such as the observation and mapping of land surfaces, including vegetation coverage (e.g., forest), geohazards, and crises [28–30]. The relatively high temporal resolution and wide scanning mode of Sentinel-1A allow for continuous time-series monitoring in the Karakoram high-altitude valley.

Landslide hazard in northern Pakistan is an increasingly catastrophic issue because of regional tectonic activity, topography, human activity, and climate change [31,32]. Focusing on the kinds of geohazards in a mountainous area such as northern Pakistan and the key core area along the KKH (Karakoram Highway), there have been several studies conducted with the construction of the China–Pakistan Economic Corridor (CPEC). These studies emphasized the growing need for landslide hazard analysis and prevention. Especially in the Hunza Valley, landslides induced by earthquakes, floods, and engineering projects have caused severe damage that has been reported and evaluated in previous research [6,8,25,31]. As the only route in the CPEC, the KKH utilized the existing old Silk Road to connect China and Pakistan and beyond. It is strategically meaningful to study the landslides in the Hunza Valley because many roads and communities are vulnerable to hazards along the KKH, and the KKH runs close to the valley floor over much of its length and is susceptible to geohazards [33]. The InSAR technique has been applied in the landslide identification and analysis along the KKH and in the Hunza Valley, and it effectively analyzed landslides. However, in the Hunza Valley, the InSAR observation was limited to applying single track Sentinel-1A data to analyze some landslides [25]. The application of both ascending and descending SAR data in this study can overcome the shortcoming of missed detection attributed to the limited observation geometry.

The present study achieved the landslide inventory updating and risk slope identification and comprehensively analyzed the landslide development and distribution mechanism in the Hunza Valley based on the SBAS-InSAR monitoring with the comprehensive application of ascending and descending Sentinel-1A and existing landslide inventory datasets, exploration of optical images, and investigation. Based on the fitting of the published data, the empirical power law relationship between area and volume was constructed and used to predict the volume of landslides in this study. Overall, this study will contribute to the advancement of landslide mapping and an improved understanding of landslide development in the Hunza Valley. This ideology and methodology can be applied to other mountainous hazard-prone segments of the China–Pakistan Karakoram Highway.

## 2. Study Area

The study area was the Hunza Valley, a key part of the Hunza Basin in the upper Indus River Basin, northern Pakistan (Figure 1). The elevation varies from 1503 m to higher than 7000 m (Figure 1c); the highest peak in the study area, Rakaposhi, is 7788 m above sea level (Figure 2a). This is a unique area hosting the highest relief and featuring multiple hazards and topographic processes in the harsh environment in northern Pakistan. Topographically, the features in this area consist of higher mountains covered by glacier ice (such as the Pisan Glacier below Rakaposhi, Figure 2b) and debris flows, and deposits of various originals such as GLOFs (glacial lake outburst floods), floods, river terraces, and slope debris flows [8,34–36] (Figure 2c). Because of the complex influence of the NW–SE-oriented faults, the region has glacially scoured valleys and abundant late Quaternary and Holocene sedimentary deposits [25,32] consisting of unsorted glacial deposits, debris flows, rock avalanches, and river gravels along the Hunza River Valley [37] that pose substantial secondary hazards (Figure 2c). The Passu, Hussaini, Gulmit, and Hoper Rivers are a major influence on landslide activity since they provide both sedimentary materials and hydrodynamic energy [33]. Supplemented by snow melting in the higher mountains, the discharge of the Hunza River increases substantially in summer, causing overbank flow and inducing a potential landslide hazard [8].

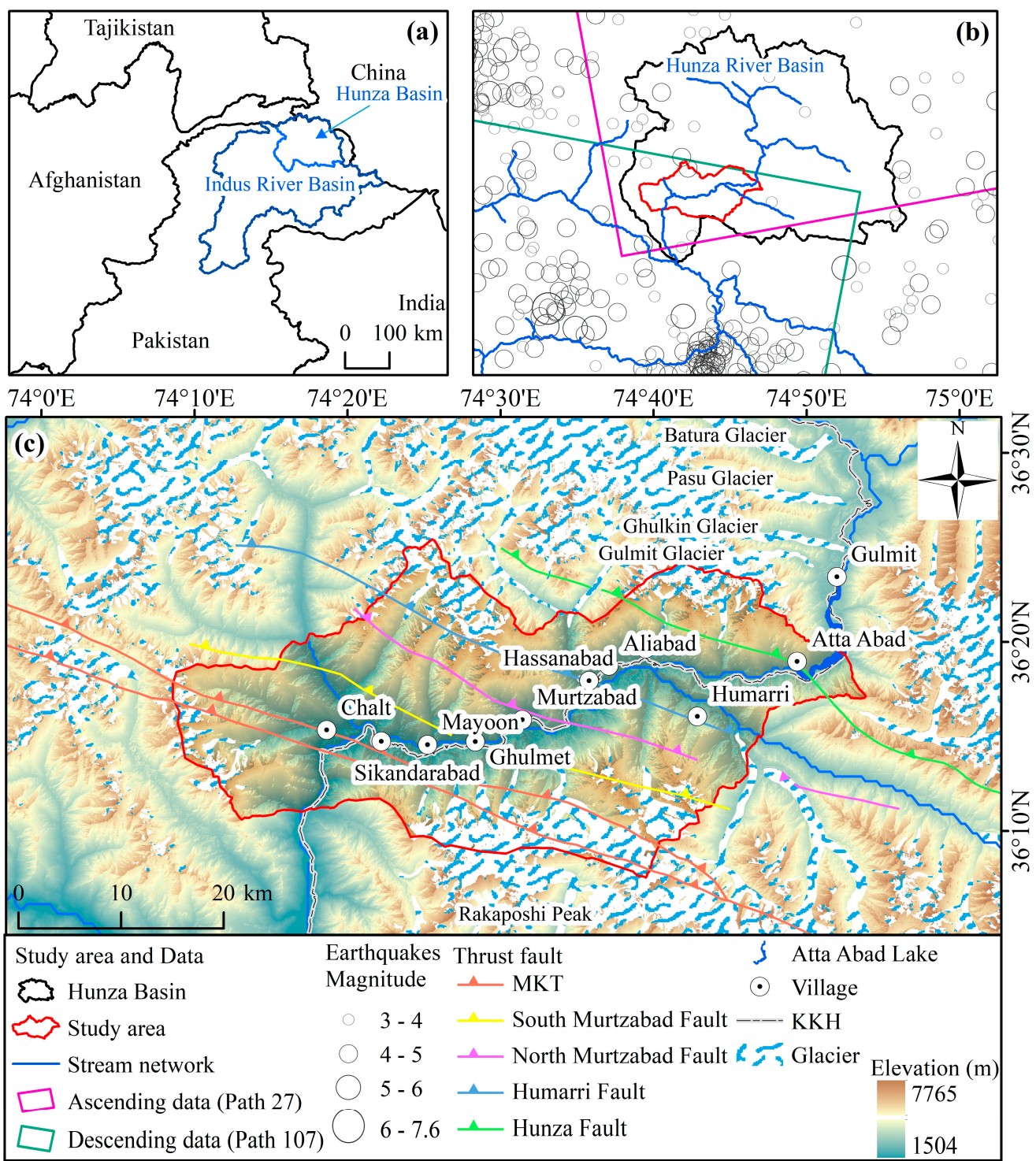

**Figure 1.** (**a**) Location of the Hunza Basin in upper Indus River Basin in North Pakistan. (**b**) Study area is represented by the red line in the Hunza River Basin, and the stream network, history of earthquake events, and the Sentinel-1A datasets in ascending and descending track are shown. (**c**) Topography environment, faults, and glacier distribution in the study area.

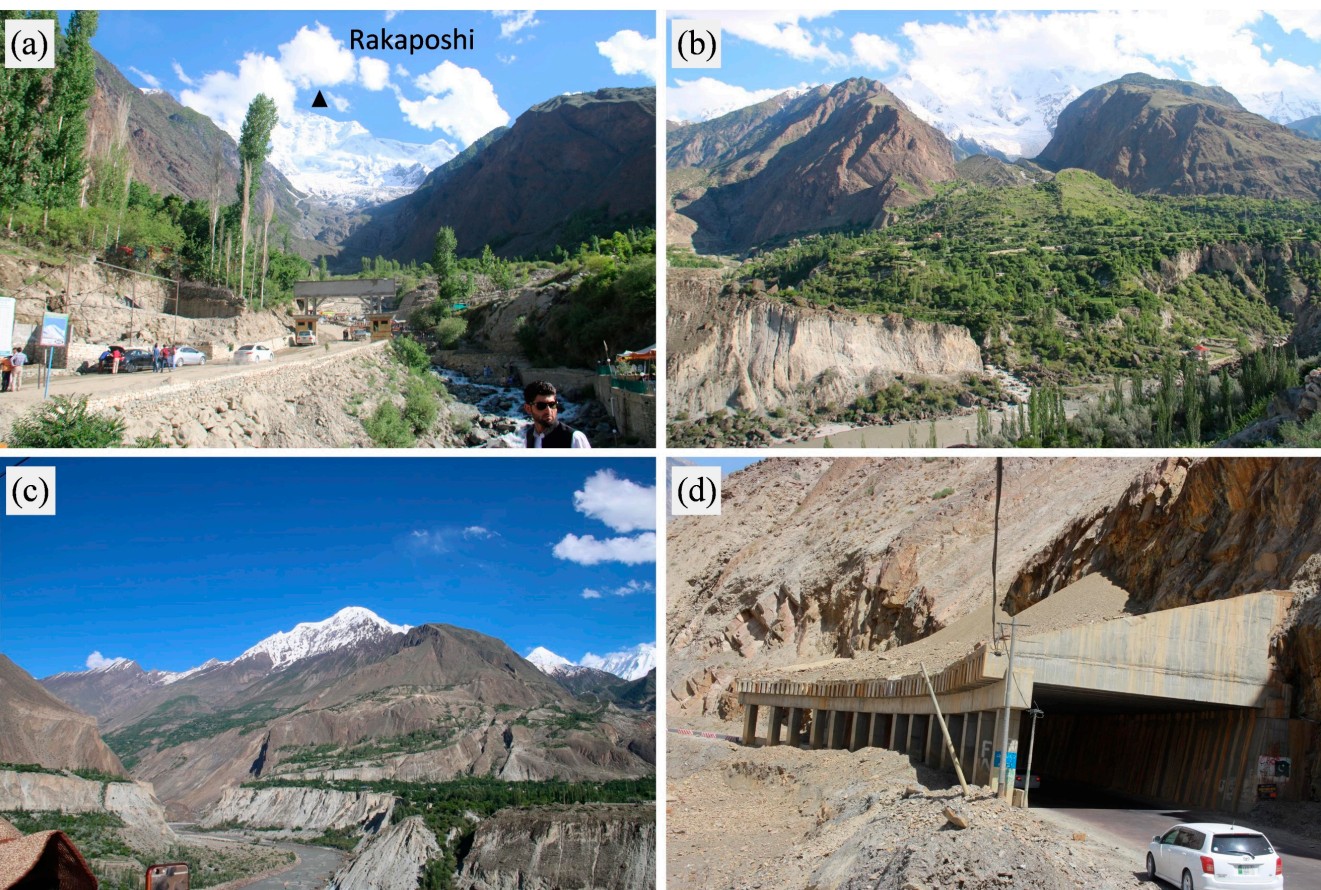

**Figure 2.** The geology environment for landslide development in Hunza Valley. (**a**) View of Rakaposhi (7788 m asl.). (**b**) The Pisan Glacier below Rakaposhi. There is intensive cultivation and quite dense rural settlement that extends from the terraced valley fill on the left bank of the Hunza River up to the middle slopes of moraines. (**c**) A geological perspective of the river terrace, glacier moraines, and landslide deposits. (**d**) South of Atta Abad, there are five open tunnels constructed to reduce the direct impact of landslides on the highway along the new KKH that was relocated because of submersion by the Atta Abad landslide dam lake.

As a result of the collision of the Indian and Asian plates, the principal tectonic expression in the Hunza area is the thrust slip between the Main Karakoram Thrust (MKT) and the Karakoram Fault [25] (Figure 1c). The complex landforms represented by the crisscrossing gullies and fragmented bedrock are evidence of active movement related to the MKT and other small faults [38]. As a zone with intensive tectonics, this area has been reported as the highest in the world in terms of frequent seismicity and earthquakes exceeding Mw 6 [39]. Therefore, the study area is heavily affected by widespread slope movement [40,41] owing to the complex impact of the intensive tectonic activity and fragile geology. Landslide hazards in the Hunza Valley are also closely related to the engineering work along the KKH, and the erosion of the Hunza River. The large landslide Atta Abad dammed the Hunza River and blocked the KKH in 2010. This prompted KKH relocation and upgrading after submersion by the Atta Abad landslide dam lake. Five open tunnels were constructed to reduce the direct impact of landslides on the highway along the new KKH (Figure 2d). Landslide hazards are frequent and have resulted in traffic disruption and severe material losses, which are major constraints on human life and socioeconomic development [7]. They therefore require intensive study.

## 3. Methodologies and Datasets

The methodologies and processing used in the present study consisted of the following steps: data preparation, displacement and velocity monitoring based on SBAS-InSAR and preliminary interpretation, field survey, validation of landslide inventory, compilation of a dataset of the latest landslides, and landslide distribution mechanism analysis. A flow chart of this methodology is illustrated in Figure 3.

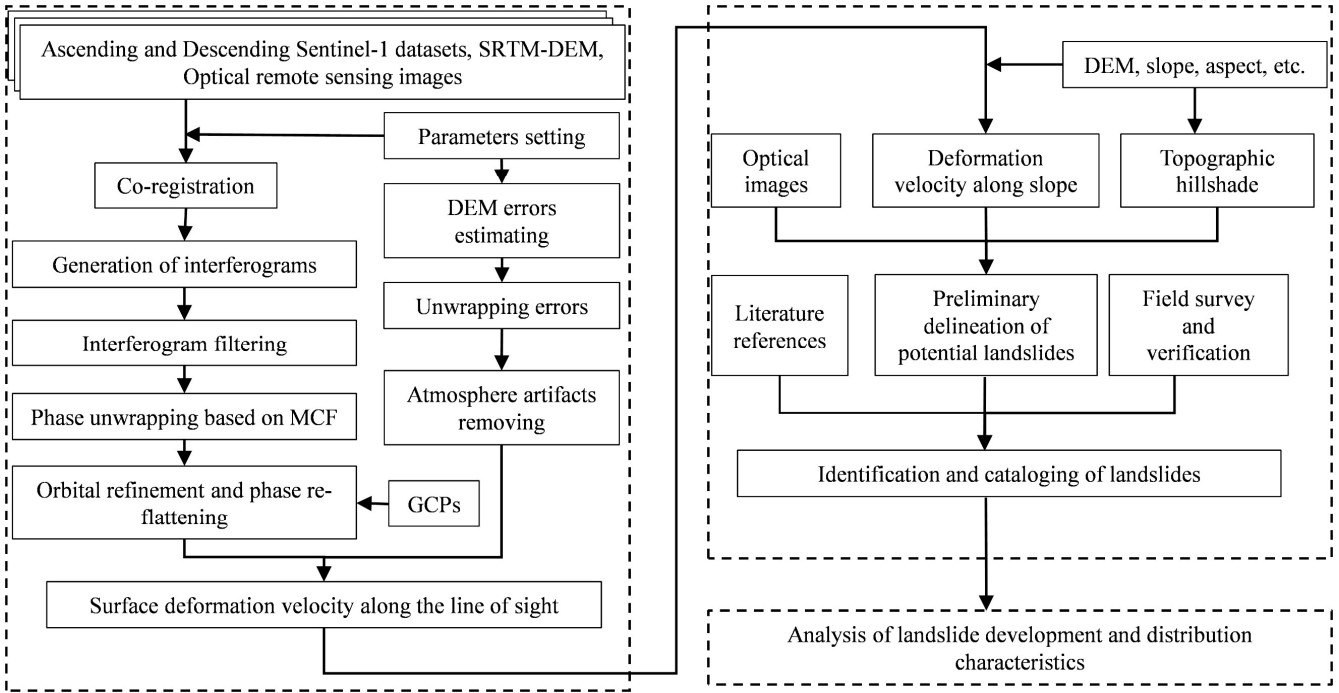

**Figure 3.** Flowchart for deformation monitoring, landslide detection, and analysis.

### 3.1. SBAS-InSAR Techniques and SAR Data

The time-series SBAS-InSAR technique with its deformation-detection ability that was proposed and developed by Berardino et al. and Lanari et al. [42–44] was applied and demonstrated in studying surface deformation and analysis in various fields, including ground subsidence detection, landslide identification, and glacier tracking [26,45–47]. Compared with PS-InSAR (Persistent Scatters Interferometry), SBAS-InSAR has a greater capability of monitoring deformations over the rugged mountainous area [48] and can provide valuable information for detecting surface deformation and time-series characteristics for analyzing the pattern and cause of active landslides [49,50].

Forty-five ascending (27) Sentinel-1A images from 2 January 2019 to 13 June 2020 and forty-two descending (107) Sentinel-1A images from 8 January 2019 to 19 June 2020, for a total of eighty-seven images, were acquired and applied to monitor slope deformation in Hunza, based on SBAS-InSAR (Table 1). Digital elevation model (DEM) data with a pixel size of 30 m, generated by the Shuttle Radar Topography Mission (SRTM), were used to eliminate the residual topographic phase for analyzing deformation results. Basic parameters of optical images and SAR datasets are listed in Table 1.

**Table 1.** Basic parameters of optical images and SAR datasets in this study.

| Satellite Mission (Source) | Sentinel-1A | Sentinel-1A | Google Earth Data |
|---|---|---|---|
| Purpose | Time-series displacement analysis | Spatial pattern analysis of displacement | Evolution precession |
| Band | C | C | Optical |
| Wavelength (cm) | 5.6 | 5.6 | - |
| Incidence of LOS (°) | 32.3–36.3 | 34.5–38.1 | - |
| Average Incidence of LOS (°) | 34.02 | 36.35 | - |
| The azimuth of LOS (°) | 83.94 | −83.94 | - |
| Path | Ascending (27) | Descending (107) | - |
| Frame | 117 | 473 | - |
| Resolution in azimuth (m) and range (m) | ~20 × 5 | ~20 × 5 | <2 |
| Minimum temporal baseline (days) | 12 | 12 | - |
| Number of images | 45 | 42 | - |
| Temporal span | 2 January 2019–13 June 2020 | 8 January 2019–19 June 2020 | 2017–2020 |
| Multi-looking (azimuth × range) | 1 × 4 | 1 × 4 | - |

In the methodology process, all images were first resampled and processed to generate interferograms with the following parameters: multi-looking factor of 4 in range and 1 in azimuth; temporal and spatial baselines of less than 70 d and 210 m, and 70 d and 200 m, respectively, for ascending and descending Sentinel-1A images [51,52] (Figure 4). Additionally, the Goldstein Filter [53,54] and the Minimum Cost Flow (MCF) algorithm [43,55] with a coherence threshold of 0.3 were used for filtering to increase the SNR and the coherence of interferograms and phase unwrapping. After the estimation of residual height and displacement information via the process of refining, re-flattening, and inversion, the atmospheric signal phase was observed and removed by the application of high-pass and low-pass filters in the temporal and spatial domains, respectively [56,57]. Singular value decomposition (SVD) was applied to estimate the deformation at each SAR acquisition date since more than one subset was available [42,43,58]. The final results enabled the detection of annual surface deformation rates in the study area.

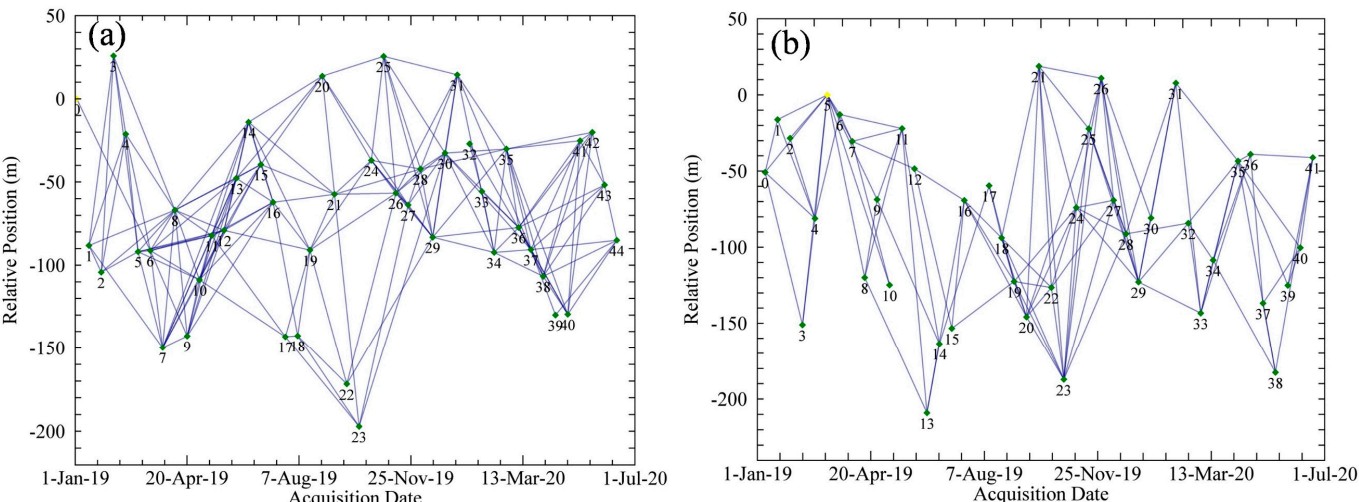

**Figure 4.** Spatiotemporal baseline map of SBAS-InSAR interferometric data of ascending track (**a**) and descending track (**b**).

### 3.2. Interpretation of Displacement Velocity and Landslides Inventory

To achieve the goals of comprehensive landslide identification and analysis in the Hunza Valley, a landslide inventory was conducted by integrating the displacement rates derived from the ascending and descending data by SBAS-InSAR monitoring, the visual interpretation of optical remote sensing images, a literature review, and field investigations.

However, in some cases, the velocity along the line of sight could not directly indicate the actual movement of slope materials. In the case of landslide detection, most landslide or ground surface deformation occurs along the direction of the slopes, and therefore, the deformation velocity along the slope ($V_{slope}$) was calculated for both ascending and descending paths with reference to the methods in the literature [23,59,60].

$$V_{slope} = \frac{V_{los}}{Index}, \tag{1}$$

$$Index = n_{los} \cdot n_{slope}, \tag{2}$$

$$n_{los} = (-sin\theta cos\alpha, sin\theta sin\alpha, \ cos\theta), \tag{3}$$

$$n_{slope} = (-sin\beta cos\varphi, -cos\beta cos\varphi, \ sin\varphi), \tag{4}$$

$$n_{slope} \cdot n_{los} = (sin\theta \cdot cos\alpha \cdot sin\beta \cdot cos\varphi) - (sin\theta \cdot sin\alpha \cdot cos\beta \cdot cos\varphi) + (cos\theta \cdot sin\varphi) \tag{5}$$

where the $V_{slope}$ is velocity along the slope, $V_{los}$ is the velocity along the line of sight. $\beta$ is the aspect of slope, $\varphi$ is the slope gradient, $\theta$ is the angle of incidence of the satellite sensor, $\alpha$ is the angle between the direction of the satellite orbit and true north (which is the radar satellite flight direction), ascending data are negative, and descending data are positive. Based on the statistics and previous research in this study area, before the conversion, we set the value in the section of ($-0.3\sim0$) as $-0.3$, and set the value in the section of ($0\sim0.3$) as $0.3$.

In addition to the optical remote sensing images and literature, the basic knowledge and imprints of the landslides in this area enabled classifying the rendering of deformation. In setting a stable threshold, the main principle is to indicate and distinguish the relative stable zone and the deformation positions using high deformation velocity. In this process, the optical image is available as supplement data to ensure the rendered higher deformation matches well with slopes experiencing displacement, as shown in Figure 3. Based on the principle explained above, this work used displacement information to identify and delineate the landslide step by step. Some of these steps have proven useful for detecting and identifying active landslides [61,62].

Firstly, an annual deformation velocity of $-20$ mm/y along the slope direction was set as the threshold for distinguishing the relative stable area and the suspected active slope in which the landslide will occur. Secondly, the preliminary mapping of suspected active landslides was conducted by superimposition on optical remote sensing images with reference to surface geomorphological features (e.g., scarps, sliding masses, and bulging toes). Finally, a field investigation was conducted to delineate the landslide mapping. In the field survey, landslide characteristics including topographic features, deformation evidence such as cracks, fissure, scarp, and the depositions were the criteria to verify the SBAS-InSAR monitored displacement information; the results will be presented in Section 4.1. In the internal work, all landslides were uniquely numbered and named.

### 3.3. Analysis of the Landslides' Development Characteristics

The landslide development is under the controlling impact of the geology tectonics and the topographic geomorphology. The landslide identified in this study was the result of various causes in the present environmental and geology conditions. These effects were reflected in the SBAS-InSAR monitored displacement information to a degree. To explore the landslide spatiotemporal characteristic, especially the active landslides detected by InSAR, a spatial statistical analysis of the detected landslides and historical landslides can

provide insight into understanding landslide hazards and their development in the Hunza Valley and Karakoram Mountains.

The topographic condition exerts an impact on the landslide by controlling the shape, slope, and position to influence the potential energy, which in turn further impacts on the hydrothermal conditions, catchment conditions, infiltration conditions, and finally mass movement processes. Differences in the topographic conditions certainly contribute to the spatial development differences of landslides. In relation to the topographic features, the landslide displacement velocity is the dependent variable and reflects the activity level and activity pattern. The deformation pattern can also be investigated using the time series information in the SBSA-InSAR results. This study explored the classes of deformation pattern and compared the characteristics of landslides from different sources.

Here, the characteristic database was established on the basis of a comprehensive landslide inventory. The landslide area, observed topographic features of elevation, relief (height), aspect, slope, and NDVI (Normalized difference vegetation index) were extracted and assigned to every landslide. The deformation rates of each individual landslide were acquired based on the SBAS-InSAR monitoring. NDVI is the reflectance of the vegetation. It is formulated as the following equation:

$$NDVI = (R_{NIR} - R_R)/(R_{NIR} + R_R) \tag{6}$$

where $R_{NIR}$ is the reflectance of the near infrared band, and $R_R$ is the reflectance of the red band. The data is derived using Landsat-8 images provided by National Aeronautics and Space Administration (NASA).

## 4. Results and Analysis

### *4.1. Ground Surface Deformations and Landslides Inventory in Hunza Valley*

#### 4.1.1. Displacement Velocity along the Line of Sight

The decorrelation in the higher region that is covered by snow, ice, and glacier limits the detection of ground surface deformation and brings more noise and uncertainty. The deformation velocity along the line of sight ($V_{los}$) from ascending and descending Sentinel-1 datasets was detected using SBAS-InSAR, setting 0.3 and 0.4 as the coherence thresholds, respectively.

The retrieved displacement $V_{los}$ was mainly concentrated nearly in the valley in Hunza, as shown in Figure 5. The negative values (red color) indicate the ground moving away from the sensor and its mean velocity in the monitoring period, and the positive values (blue color) represent the displacement toward the satellite sensor and its mean velocity. There are points with velocity values higher than 20 mm/y on the valley runout and toe of the slope, which may indicate the process of the sliding. Regarding the density of the coherent targets, 551,048 and 697,777 coherent targets points were obtained from the ascending (Figure 5a) and descending Sentinel-1 (Figure 5b) datasets, respectively. The densities of coherent points were calculated as 439 points and 557 points per square kilometer in the study area with an area of 1252.6 km$^2$ for the ascending and descending datasets, and the detected displacement information covered the main part of the valley below the elevation of 3500 m (Figure 5). Because of topography conditions such as an extremely steep slope and environmental limitations of the glacier, ice, and dense vegetation, the Sentinel-1 datasets had lower sensitivity over the higher alpine area, resulting in the limited availability of coherent targets in these areas.

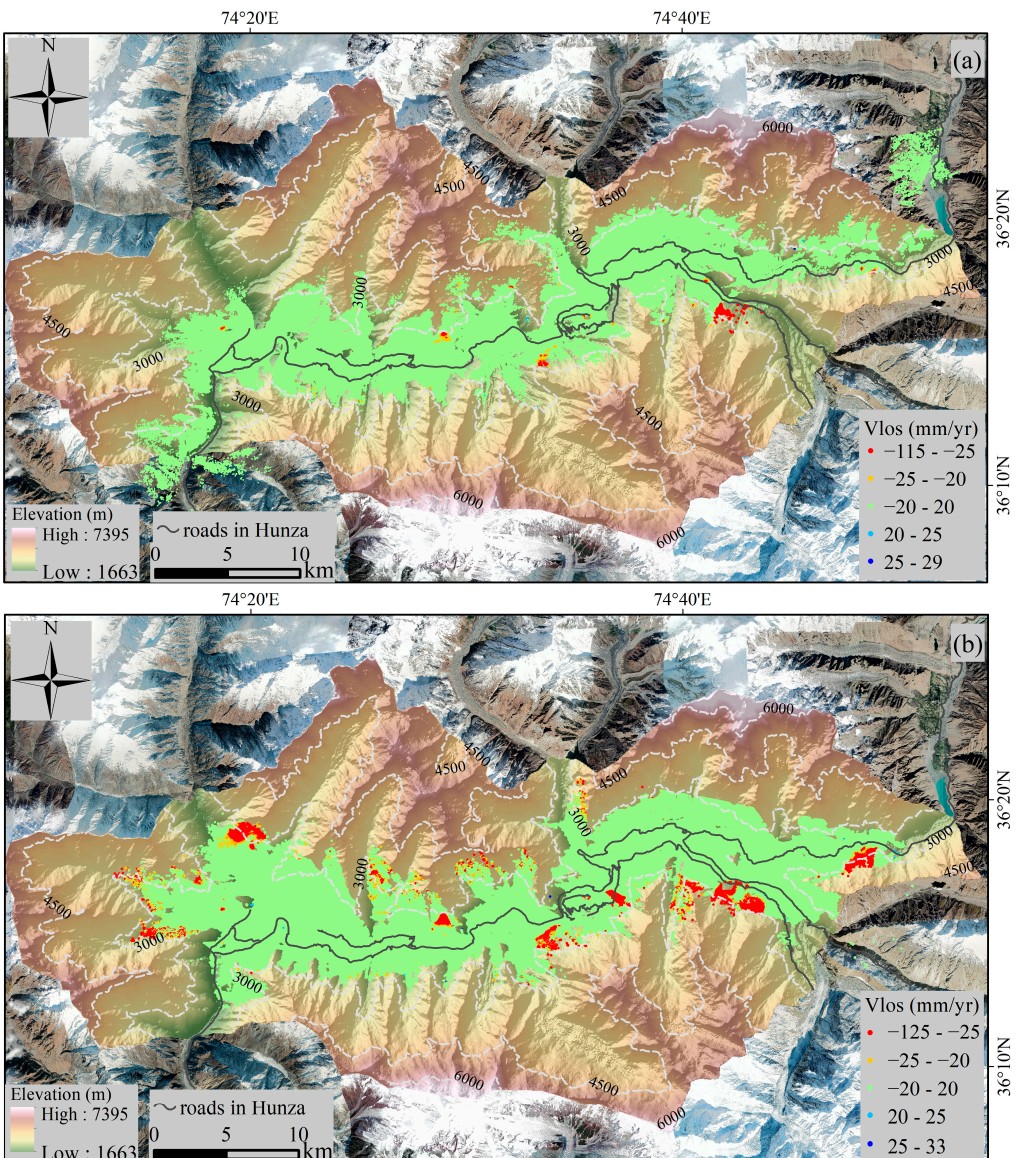

**Figure 5.** Displacement velocity along the line of sight (V$_{los}$) monitored by SBAS-InSAR applying the ascending (Path 27, Frame 117, in (**a**)) and descending orbit data (Path 107, Frame 473, in (**b**)) Sentinel-1A dataset. The black line represents the roads in the Hunza Valley.

4.1.2. Displacement Velocity along the Direction of Slope

The simulation of the displacement velocity along the slope facilitated landslide identification by providing a direct reference for deformation investigation, based on the method in Section 3.2.

Figure 6 shows the results of displacement velocity along the slope (V$_{slope}$) acquired from the ascending and descending Sentinel-1 datasets. After conducting the classifying rendering, referring to the optical image, this study set the velocity threshold for landslide identification. The maximum value of ground surface displacement velocity along the slope was 311 mm/y and −490 mm/y, which were monitored by the ascending and descending Sentinel-1A datasets (Figure 6). There were 109,687 of 118,315 (92.7%) detected coherent points within the deformation interval of (0, −20) for the ascending dataset and 109,153 of 139,699 (78.13%) detected coherent points within the interval of (0, −20) for the descending dataset. This indicated that most of the study area was stable.

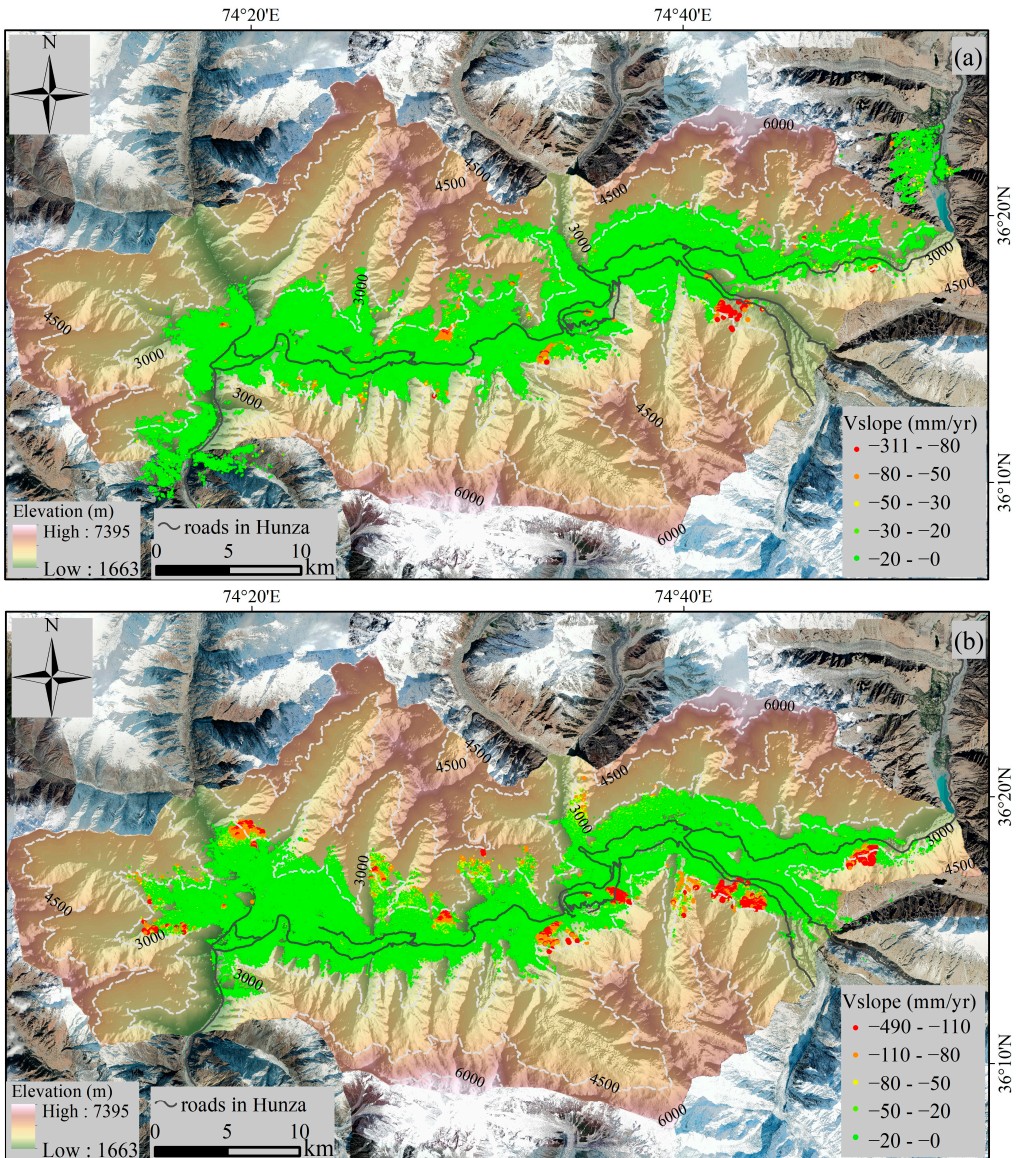

**Figure 6.** Displacement velocity along slope and landslides detected by SBAS-InSAR applying the ascending (Path 27, Frame 117, in (**a**)) and descending orbit data (Path 107, Frame 473, in (**b**)) Sentinel-1A dataset. The black line represents the roads in the Hunza Valley.

### 4.1.3. Landslides Inventory

Combined with the displacement velocity along the slope derived from the ascending and descending datasets ($V_{slope}$), the visual interpretation of optical remote sensing images and field investigations were conducted. First, the interpretation of the velocity along the slope and optical images ensured the preliminary delineation of the boundary by referring to deformation velocity with relatively high values, topographic characteristics (obtained from DEM), and the optical image features. These slopes were the suspicious objects prepared for the field investigation and validation. Moreover, the deformation area and position on the slope, the scarps and fissures, and the boundary of the potential landslide failure were the items that needed to be verified. Finally, any one slope that could be acknowledged as consistent with the monitoring results was inventoried in the field. From the survey, these landslides in the Hunza Valley were characterized by round-backed, armchair-like back walls, cracks, fragmented rocks, and fragment deposition; additionally, the absence or near-absence of vegetation cover on the talus deposits was also evidence of active rock falls.

Consequently, 53 potential landslides with a total area of 48.8 km² were detected and identified based on the SBAS-InSAR detected deformation velocity. In addition, 65 landslides were gleaned and delineated by referring to the literature [6,25,35] and the remote sensing image combined with field investigation (Figure 7). We assigned unique numbers and names to all landslides. Based on the inventory, we summarized morphological features such as an irregular ellipse, long tongue, long rectangle, and wide rectangle. The area of landslides ranged from 0.05 to 6.4 km². In the 53 SBAS-InSAR detected landslides, there were 20 landslides exclusively detected by ascending Sentinel-1 datasets and 28 detected especially by the descending Sentinel-1 datasets; the remaining 5 landslides were detectable by both ascending and descending Sentinel-1. These findings indicated that the combined application of ascending and descending data can overcome the limitations of the acquisition geometry of a single scanning posture.

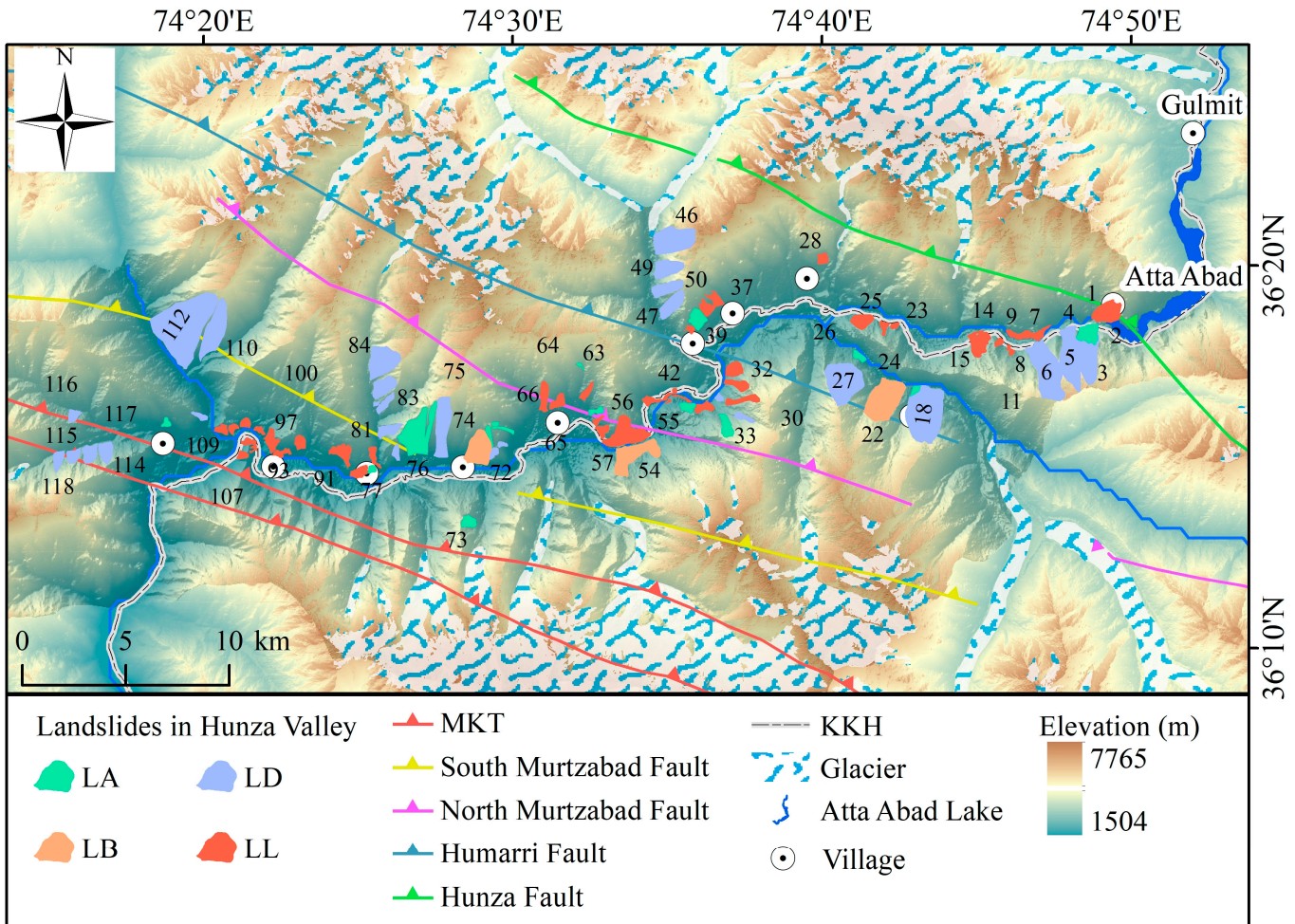

**Figure 7.** Distribution of landslides detected with multi-track Sentinel-1A data based on SBAS-InSAR, and thrust faults and glaciers in the Hunza Valley, northern Pakistan. LA: the landslide detected by ascending Sentinel-1A, LD: the landslide detected by descending Sentinel-1A, LB: the landslide detected by both ascending and descending Sentinel-1A, LL: the landslide cited from literature.

In the detection and identification of landslides, the SBAS-InSAR detected deformation and the features of images and field photograph were useful in combination. Figure 8 comparatively shows typical examples of detected landslides and two landslides that were interpreted based on images and field investigation in the inventory. These landslides were mostly covered by SBSA-InSAR-detected coherent targets. The first four landslides were the active landslides identified based on the SBAS-InSAR monitoring (Figure 8a–f). Because of the steeper slope or the actual inactive state, some landslides may not have strictly

corresponded to the criteria of higher deformation velocity for identifying landslide and experienced lower rates of deformation. These landslides were experiencing deformation and occurred previously (in history) or had developed after the ancient events (Figure 8g,h). These landslides were delineated through the field investigation and optical remote sensing image interpretation in combination with referral to the literature (Figure 8g,h).

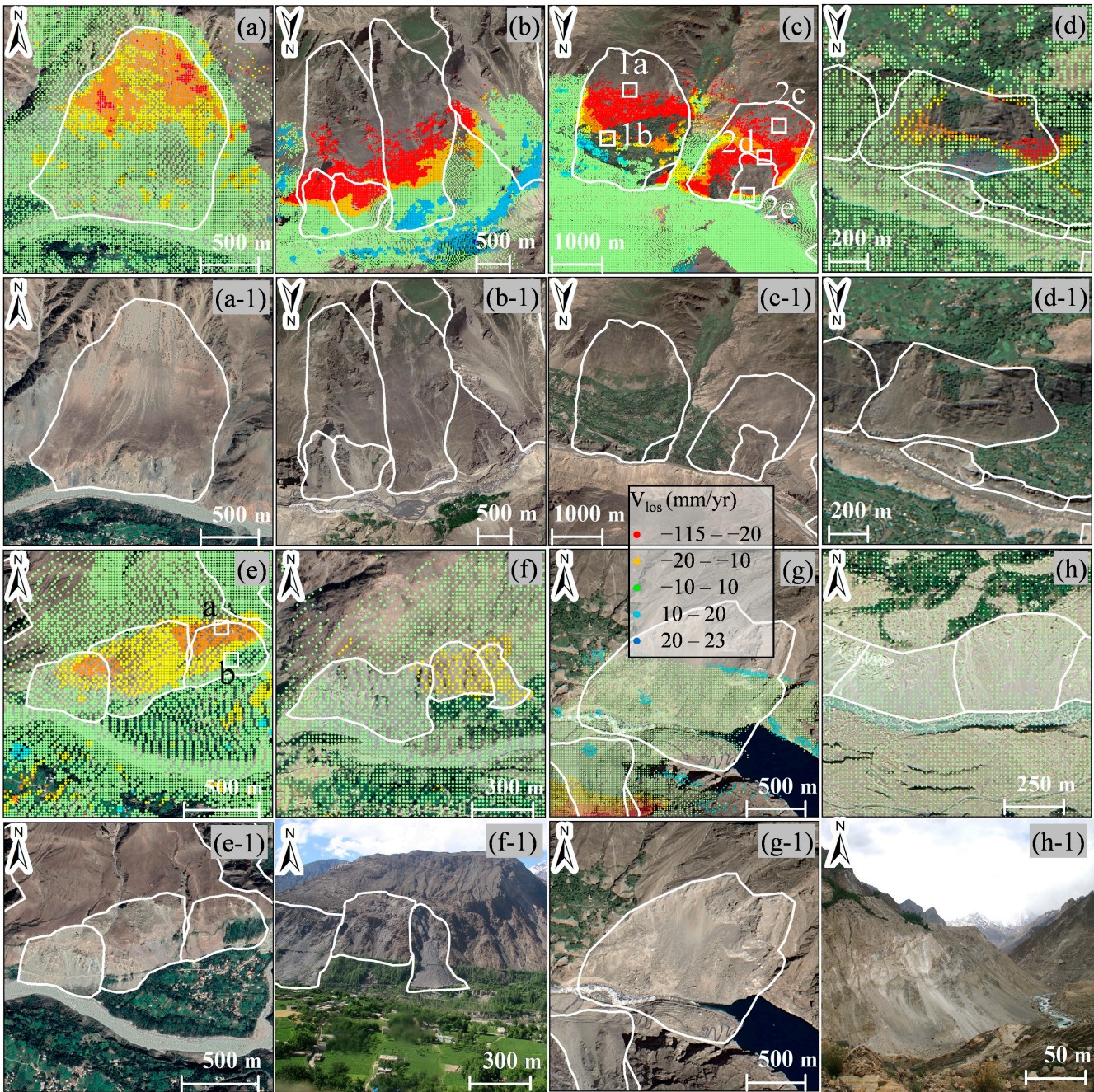

**Figure 8.** SBAS-INSAR monitoring deformation and the optical image interpretation of identified (**a**–**f**) and interpreted (**g**,**h**) landslides. (**a**): No. 72 Ghulmet landslide, (**b**) No. 2–6 landslides, (**c**) No. 18, 22 Humarri-1, Humarri-2 landslide, (**d**) No. 42 landslide, (**e**) No. 87-89 Mayoon-1~3 landslides, (**f**) No. 94-96 Jizalabad landslides, (**g**) No. 1 Atta Abad landslide, (**h**) No. 9-10 landslides.

*4.2. Spatiotemporal Characteristics of Landslides in Hunza Valley*

4.2.1. Development Characteristics of Landslides

Based on the comprehensive, updated landslide inventory, statistical analysis of spatial topographic, deformation, and other attributes of the landslides from different sources can improve our understanding of the spatial–topographic developments of landslides and their related inner causal mechanism in the Hunza Valley. For the landslides interpreted through images and investigation, the literature also provided useful information. Here, we identified them as landslides as cited in the literature. To inquire about the detailed characteristics, the detected landslides and landslides cited in the literature were plotted in box plots and in the polar diagram. The landslides' attributes including $V_{slope}$, area, volume, max elevation (at the top point of the landslide), height (difference between the maximum elevation and the minimum elevation), aspect, slope, and NDVI (normalized difference vegetation index) were investigated.

(1)    Deformation pattern and velocity

The SBAS-InSAR derived deformation velocity is valuable when delineating the boundary of an active landslide. In addition, the time series analysis of deformation in the monitoring period is useful for investigating the landslide development pattern. Previous studies have proposed that the statue of deformation curve is of value when indicating the pattern of landslide development, and the extra high acceleration is a criterion of the possibility of failure [63–65]. It is vital to explore the deformation pattern in the monitoring period not only to gain the knowledge of potential landslides but also to choose typical objects to study specifically for risk management. The displacement trends of each landslide were checked using the time series displacement data and, considering geomorphological features observed in the field and on images, were classified manually to ensure accuracy. In this study, the trends of the landslide deformation process were classified into constant rate (Figure 9a), accelerating (Figure 9b), decelerating (Figure 9c), and fluctuating (Figure 9d). Of these landslides detected by SBAS-InSAR, 28 experienced constant rate deformation, 20 experienced accelerating deformation, and 5 experienced decelerating deformation (Figure 9e). On the other hand, the interpreted landslides experiencing constant rate, accelerating, decelerating, and fluctuating were 31, 11, 4, and 19, respectively (Figure 9e). It was determined that the majority of detected landslides (90%) had experienced displacement with constant rate or accelerating rate, and the majority of landslide supplied from the literature (77%) had experienced displacement with constant or fluctuating rate (Figure 9e). It was clear that the landslides interpreted from the literature experienced steady deformation or irregular surface movement, demonstrating that the detected landslides were more likely to develop into accelerating rate and had an approximate risk of failure in the future [35].

The landslides identified by the Sentinel-1A dataset (LA, LD, and LB) were characterized by higher displacement velocities compared to the landslides derived from the literature (LL) (Figure 9f). On the contrary, the landslides interpreted from the optical images from the literature had unique features of lower velocity and smaller area (Figures 9f and 10a). This was the reason that these smaller landslides were excluded in the SBAS-InSAR-based landslide detection and identification: because they were relatively stable with extra local deformation or extra slow-moving velocity and smaller area. This reflected the purpose in basing the research on SBAS-InSAR when updating the landslide inventory, and the literature review aided as the significant supplementary work.

(2)    Magnitude: area and volume

The landslide areas possessed unique differences among the four different resources (Figure 10a). The areas of the landslides detected by descending data (LD) were obviously greater than others and had a larger range of 0.06 km$^2$ to the largest of 6.4 km$^2$. On the contrary, the largest area of the landslides detected by ascending dataset (LA) was 2.21 km$^2$, and largest area of the landslides cited from the literature (LL) was 1.16 km$^2$ (Figure 10a). The 42.39 km$^2$ area of the 33 descending detected landslides accounted for 67% of the area

of all sources of landslides (63.23 km$^2$) and 86.7% of SBAS-InSAR-detected landslides by Sentinel-1A datasets (48.8 km$^2$). On the whole, the area of landslides in this study was in the range of $2.5 \times 10^4$ m$^2 \leq A_L \leq 6.4 \times 10^6$ m$^2$.

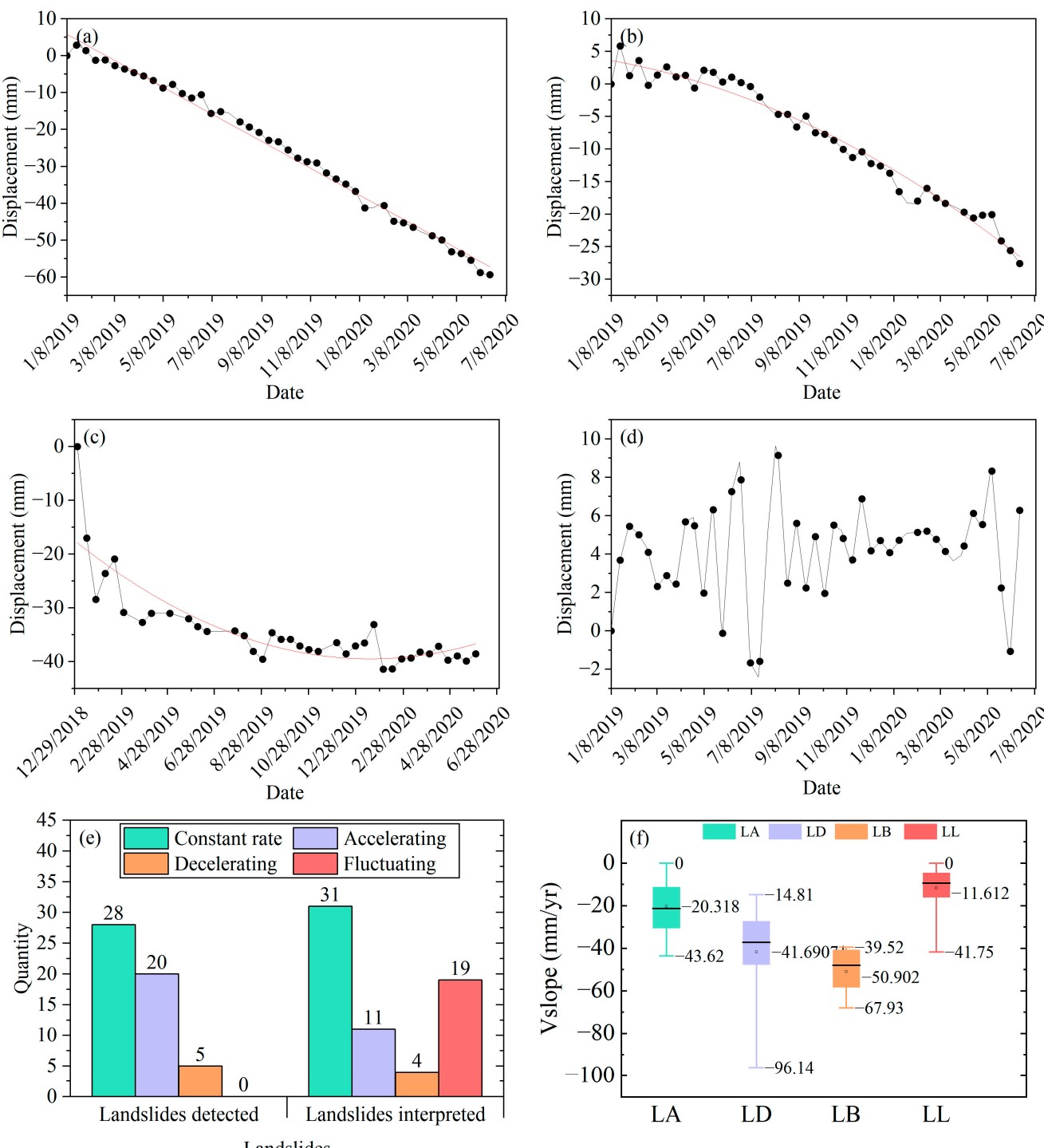

**Figure 9.** Typical time-series of four types of pre-failure strain. (**a**) Constant rate, (**b**) accelerating, (**c**) decelerating, and (**d**) fluctuating strains; (**e**) statistical quantity of landslides in different pre-failure strains; and (**f**) multi-source landslides development characteristics of velocity, the landslide statistic includes landslides detected by ascending (LA), descending (LD), both ascending and descending Sentinel-1A datasets (LB) and the landslides from literature (LL).

To model the empirical relationship between $A_L$ and $V_L$, a distinct power law was fitted to the published data [56]:

$$V_L = a \times A_L{}^\alpha \tag{7}$$

where $A_L$ (m$^2$) is the area of landslide, $V_L$ (m$^3$) is the volume of landslide from the published dataset [66], a and $\alpha$ are curve-fitting parameters to link landslide area ($A_L$) to landslide volume ($V_L$) based on recent research [67–69]. The model fitted based on the published data of landslides in the Hunza Valley was relevant to a broad range of landslide areas ($2.426 \times 10^3$ m$^2$ < $A_L$ < $1.6 \times 10^6$ m$^2$) [56] (Figure 11). The large landslide area of the reported data covered same orders of magnitude as this study, and the maximum magnitude was $10^6$. We found that SBAS-InSAR-detected landslides were larger in volume compared to the landslides interpreted by referring to the literature (LL) (Figure 10b). The minimum magnitude of fitted data was relatively smaller than that in this study, and the forecast revealed that the small and medium scale landslides had good prediction effect. For example, the Humarri-1 landslide was predicted with a volume of $4.7 \times 10^7$ m$^3$, which matched well with that estimated to be a large fault controlled landslide with a volume of $4.1 \times 10^7$ m$^3$ [25] (Table 2). According to the published data and the data predicted in this study, their values were similar and of the same magnitude (Table 2).

**Table 2.** Comparison of the landslide areas and predicted volumes in this study with published data [25,66].

| Landslide Name | Forecast in This Study | | Published Data | |
|---|---|---|---|---|
| | Area (10$^4$ m$^2$) | Volume (10$^4$ m$^3$) | Area (10$^4$ m$^2$) | Volume (10$^4$ m$^3$) |
| Karimabad-4 | 61.801 | 446.384 | 45.225 | 135.675 |
| Karimabad-3 | 20.383 | 99.747 | 17.105 | 153.948 |
| Miachar-3 | 43.888 | 281.117 | 38.851 | 194.257 |
| Karimabad | 25.476 | 134.821 | 38.492 | 230.956 |
| Humarri-1 | 353.527 | 4709.713 | 340 | 4100 |

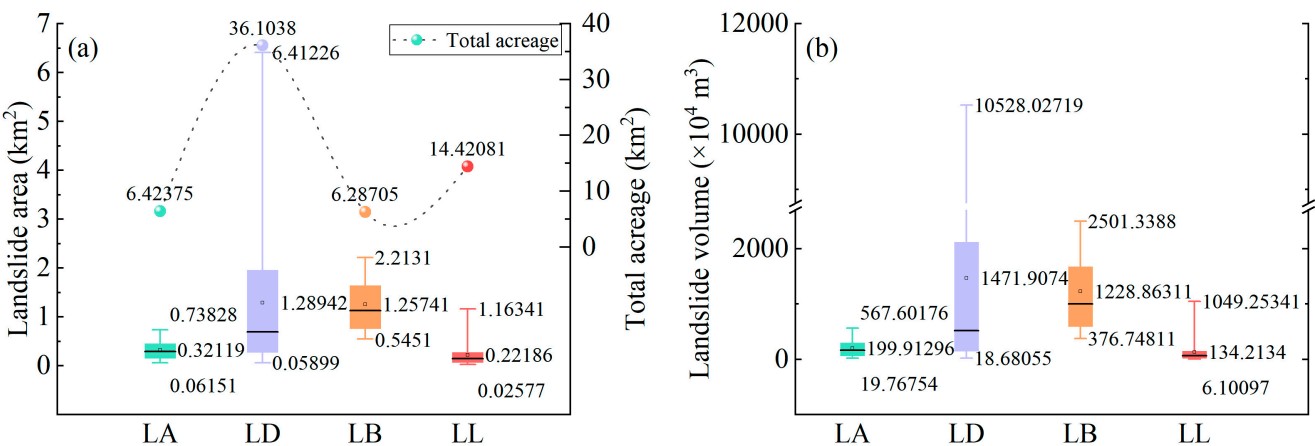

**Figure 10.** Analysis of multi-source landslide development characteristics of (**a**) area and (**b**) predicted volume.

(3) Topography

There was a topography priority condition for landslide development in Hunza Valley. The ranges of the elevation and height of the landslides detected from descending data were broader than those of the landslides from ascending datasets (Figure 12a,b). The maximum elevation of the landslides detected by the descending datasets (LD) ranged from 2261 to 4583 m. This range was broader than that of the landslides detected by the ascending datasets (LA), which ranged from 2245 to 3803 m. The point is that the landslides cited from the literature (LL) had a relatively lower maximum elevation in the interval of

2023 to 3674 m (Figure 12a). The height of landslides also exhibited similar characteristics to the maximum elevation (Figure 12b). The heights of the landslides detected by the descending datasets (LD) had border ranges from 245 to 2128 m, compared to those of the LA (215–1808 m) and LL (52–1278 m) (Figure 12b). This study claims that the Sentinel datasets have a great potential in the comprehensive detection of landslides with high altitude and high position. The Sentinel data from the descending track particularly detected more high position huge landslides in the Hunza Valley.

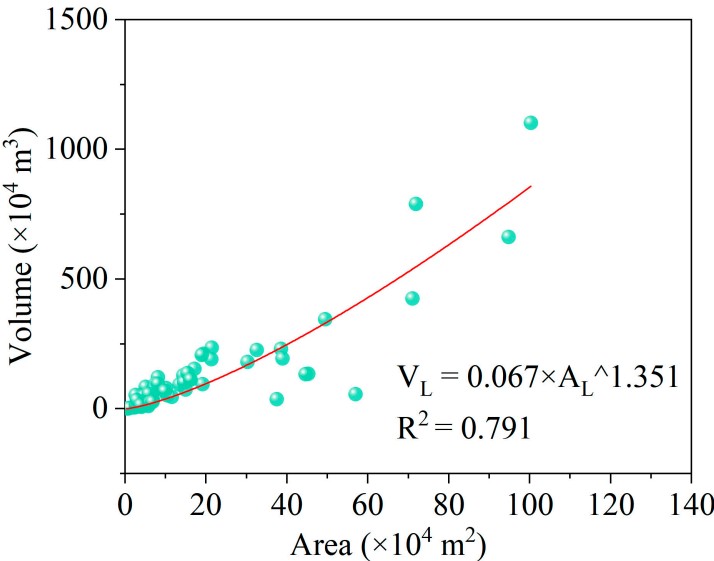

**Figure 11.** Empirical measurements for landslides of the slide type obtained through a literature search. Dots portray the area, $A_L$ (*x*-axis, $m^2$), and volume, $V_L$ (*y*-axis, $m^3$), of 72 landslides.

The spatial aspect distribution features revealed that the aspects from south to west were the prior aspects of the landslides (Figure 12c,f). In this directional range, the statistical analysis of the number and accumulating areas of landslides revealed the aggregation phenomenon of landslides. There were 76 of 118 landslides in this superior condition. Consequently, and accordingly, the sum area of these landslides was predominant at 43.75 $km^2$. That accounted for 69% of the 63.23 $km^2$ of total landslide area. It was also revealed that the landslides cited in the literature (LL) had a border aspect distribution of 20–331° (Figure 12c) and a border slope distribution of 18–47° (Figure 12d), but these landslides were predominantly smaller landslides with lower area, velocity, elevation, and height. The comprehensive application of ascending and descending Sentinel data detected landslides with a slope gradient of 26–46.7° in the Hunza Valley (Figure 12d).

(4)　NDVI

The NDVI (normalized difference vegetation index) attribution is one common substitute for land use and land cover. It was revealed that the landslides detected by the Sentinel had the NDVI attribution of 0.07–0.6. In contrast, the landslides cited in the literature (LL) were characterized by a lower NDVI. The majority of LA, LD, and LB landslides had higher NDVI attributions than LL landslides (Figure 12e). That matched well with a former study revealing that the landslides mostly developed in the bare slope [35,70].

In summary, the landslides from the literature (LL), which were interpreted from the optical image cited in the literature reference, had lower velocity, area, elevation, and height. The landslides detected by Sentinel (LA, LD, and LB) had special attributions of higher area, velocity, elevation, height, slope, and NDVI. This clarified that the reason for the disappearance of some landslides in the InSAR detection was lower velocity, and the InSAR monitoring detected landslides that were characterized by high altitude, high position, and concealment by high vegetation cover.

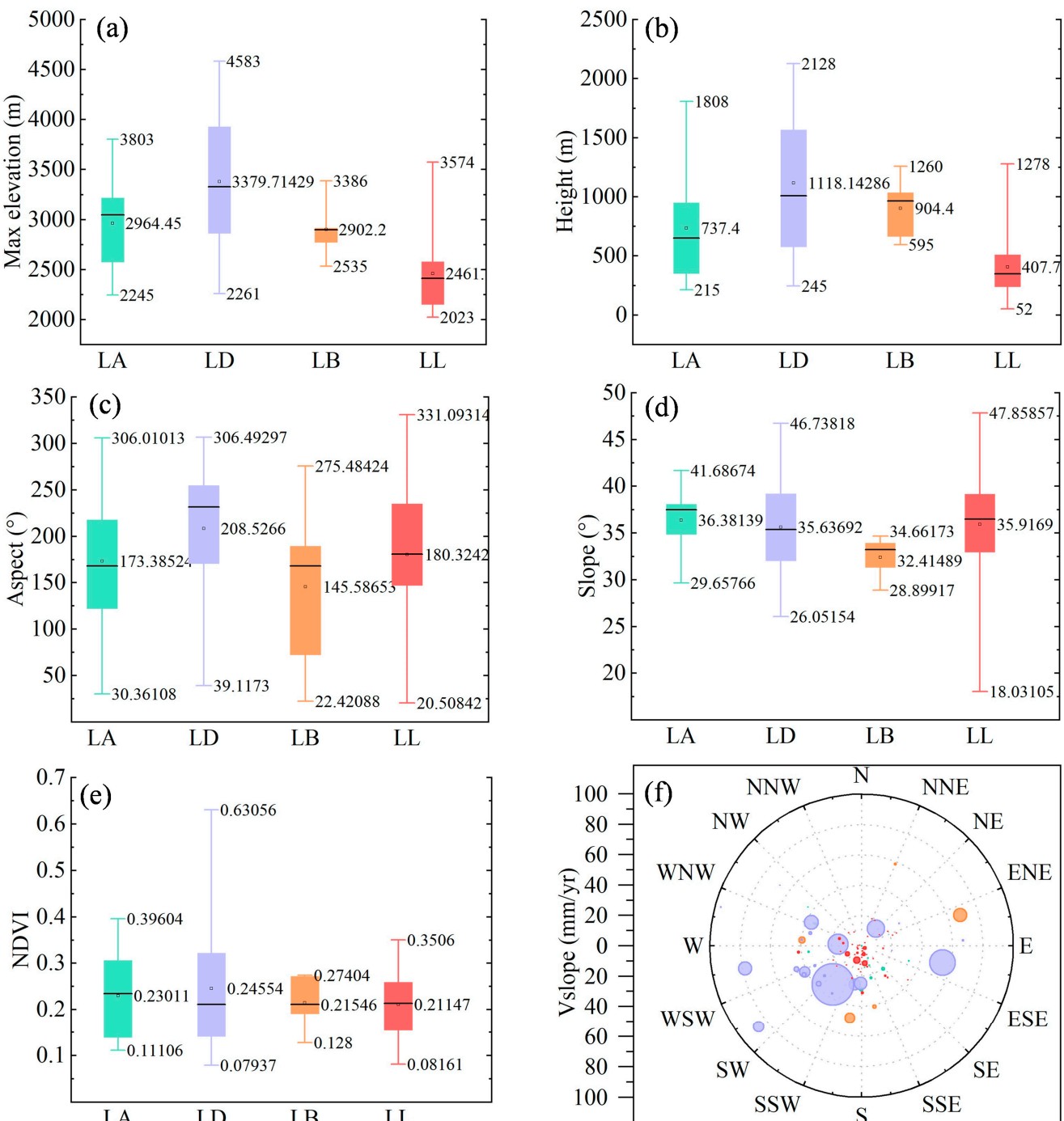

**Figure 12.** Analysis of multi-source landslide development characteristics of (**a**) elevation, (**b**) height, (**c**) slope aspect, (**d**) slope, and (**e**) NDVI, and (**f**) the colorful dots of different sizes indicate the landslide sources and their area. Its quadrant and radial locations represent the aspect and displacement velocity. The landslide statistics include landslides detected by ascending (LA), descending (LD), both ascending and descending datasets (LB) and the landslides from the literature (LL). The max–min–mean are marked on the graph.

4.2.2. Development of Active Large Landslides

Active faults have more significant control over active large landslides. Based on this investigation, the Atta Abad, Humarri-1 and 2, Mayoon, and Jizalabad landslides are typical examples of large active landslides (Figures 8 and 13). They are labeled 1, 18, 87, and 96, respectively, in the inventory shown in Figure 7. These three landslides are in close

proximity to populated villages, KKH, and the Hunza River, and they coincidentally experienced high displacement velocity in different positions during and before the monitoring period. The Atta Abad landslide dammed the Hunza River for 21 km with a 120 m high and nearly 1 km wide barrier in 2010. The landslide deposits comprised colluvial material of a fine, sandy matrix, with blocks of granite and granodiorite [8]. Although controlled by opening the dam and dredge drainage, the dam and reservoir pose the risk of a dammed lake outburst flood hazard chain to the settlements downstream (Figure 13a). A previous study indicated that the local Humarri government was completely absent and failed to identify physical evidence of potential landslides [71]. Consequently, all the households surveyed in the Humarri community showed high institutional vulnerability [72]. According to a holistic multidimensional assessment, the Humarri community was exceedingly vulnerable to hazards, especially in six dimensions [72].

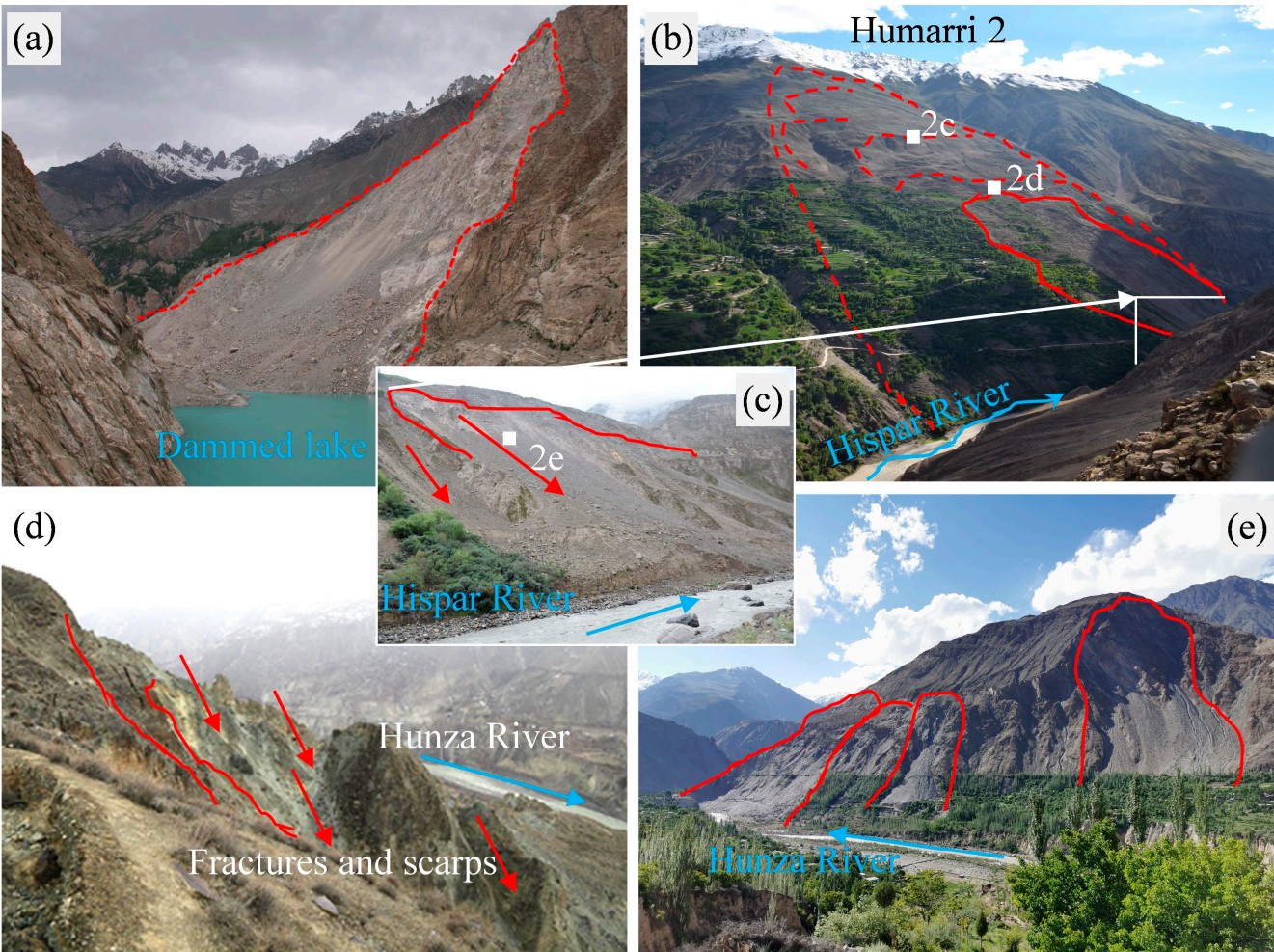

**Figure 13.** Photographs of a typical large landslide in the Hunza Valley controlled by the faults and river erosion. (**a**) Atta Abad. (**b**,**c**) Humarri-2 and secondary landslide, toe of Humarri. (**d**) Mayoon. (**e**) Jizalabad.

The Humarri village is situated on the slope composed of glacial–fluvial deposits that are underlain by high-grade metamorphic rocks of Karakoram block, on the left bank of the Hispar River (Figure 13b,c). There are two landslides at Humarri, named as Humarri-1 and Humarri-2, larger than the Atta Abad landslide, and it experiences displacements on the entire slope unit owing to it being controlled by the faults and affected by river erosion and human activities. The Huamrri-1 landslide has an elevation from 3519 m to 2127 m and a north-east sliding direction, with a length of 2.4 km and a width of 1.8 km.

The Humarri-2 landslide has a similar sliding direction and an elevation from 3386 m to 2126 m, with a length of 2.3 km and width of 1.5 km. According to the field survey, the tensile fractures and scarps were developed over the upper part, and the boundary can be delineated based on the fissures, which was the evidence for the active sliding (Figure 13b,c). SBAS-InSAR monitoring determined the landslide displacement velocity along the slope direction, which is higher than 100 mm/y on the upper parts, and revealed that the lower part experienced lower deformations soon after the activity of the upper part. The time series displacement analysis at the positions of Humarri-1 and Humarri-2 landslides showed that the upper parts experienced persistent deformation and had higher deformation rates during the monitoring period. The lower part experienced relatively slow deformation (Figure 14a,b).

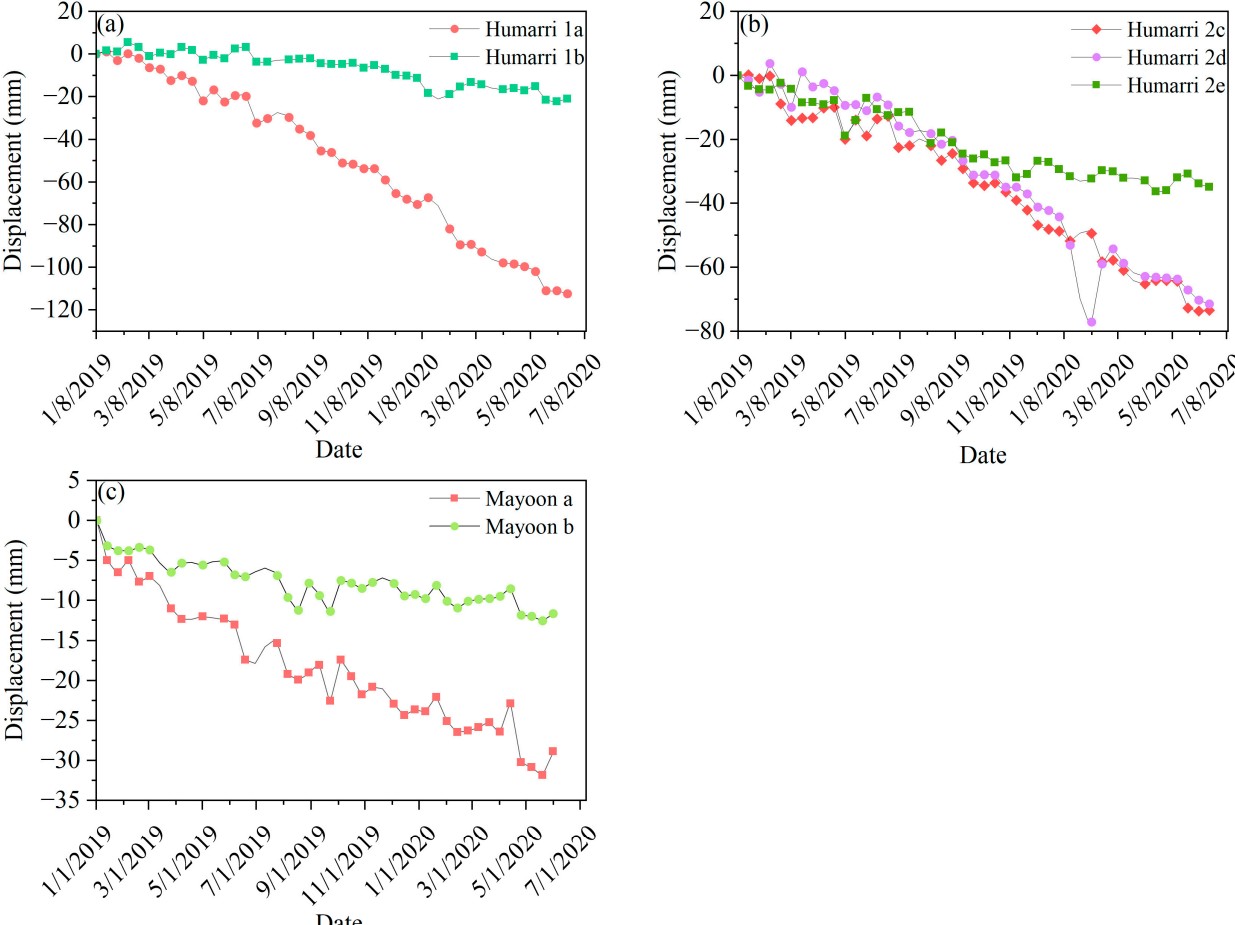

**Figure 14.** The deformation time series of large landslides. (**a**) Humarri-1; (**b**) Humarri-2; the analysis points a–e are identified in Figure 8c; (**c**) Mayoon; the analysis points a and b are identified in Figure 8f.

Tectonically, the Mayoon and Jizalabad landslides lie at the northern margin of the Kohistan Island Arc (KIA), bounded between the Main Karakoram Thrust (MKT) and the South Murtzabad Fault. The present investigation and the previous GPR tests declared that there are highly heterogeneous materials deposited in the top layers of Mayoon [5]. The head and body of the Mayoon landslide are covered by unconsolidated material and have fractures of varying lengths and widths (Figure 13d), and the fragmented and accumulated materials are obviously from the Jizalabad landslide (Figure 13e). The time series analysis revealed that the displacement histories of the Mayoon upper and lower parts are similar, but the upper part experienced larger accumulative deformations, three times greater than the lower body during the monitoring period (Figure 14c).

In summary, regarding the development mechanism of the landslide, the Humarri fault passes through the back wall of the Humarri slope over the ridge, and the South Murtzabad fault passes the ridge over the Mayoon landslide. The faults serve as controllers of the top limit on the slope and contribute to the sources of the kinetics and materials by cutting through mountains and breaking rocks. Structural uplift combined with erosion incision plays an important role in the shaping of steep slope landforms. The glacier denudation and the river erosion created a developed free face on the toe of the Humarri, Mayoon, and Jizalabad landslides. Thus, the river incised fluently, interacting with glacial debris flows. For the deformation pattern, the melting glacier water and rainfall infiltration from cracks on the upper part may have promoted the pushing action from the gravity of the upper part. Simultaneously, the coupled comprehensive actions from river erosion and active tectonics could have impacted on the stability of the slope toe.

### 4.3. Distribution Pattern and Causes of Landslides in Hunza Valley

The spatial distribution of the active slow-moving landslides in the Hunza Valley has a unique elevation zonality. Based on the average level of min elevation (at the toe) and max elevation (at the crown), the compiled landslides in this study mainly occurred in the areas with elevation between 1997 and 3379 m, indicating that the landslides mostly developed in the near valley of the Hunza River. Exploring the reasons, the distribution characteristics of landslides in the Hunza Valley are coincidently limited by the elevation of 3525 m given by Hassan in 2021 [36], above which the permafrost is distributed, and this elevation level is near to the zero-degree isotherm of 3400 m in Hunza reported by Hasson in 2014 [73]. In summary, along with the landslide development characteristics related to the terrain factors indicated in the previous chapter, the landslide distribution has a zonality of elevation. Because of the distribution of the permafrost in high-altitude areas, the slope rock and soil mass are less weathered and eroded, and the material transport process is infrequent or insignificant. Therefore, land sliding with monitorable deformation is less developed in the higher altitude area in the Hunza Valley. As to the reason for the clustering of landslides in this elevation interval of 1997 to 3379 m, except for the near proximity of river erosion and fault activities, the terrain in this section has less snow cover, is infiltrated by high-altitude meltwater, and experiences the process of seasonal freeze–thaw, which means theses slopes are more likely to experience weathering and failure.

A series of parallel tectonic faults dominated by the MKT controls the distribution and development pattern of the landslides. Complicated geology conditions controlled by the thrust tectonic activities from the Indian Plate to the Eurasian Plate contributed to the formation of a series of active thrust faults in northern Pakistan, especially in the Hunza Valley. The Hunza Valley features a dynamic geology background and various landforms and terrains including alpine landform, glacier, debris flows, terrace, and alluvial-platform and fan. The MKT is predominantly thrust and is responsible for brittle deformation in this area. Owing to the tectonic activity around the MKT and a series of smaller thrust faults, many earthquake activities have occurred in this area, and the long-term effect of seismic forces has made this area prone to landslide hazards [8]. It is clear that the detected active landslides, especially the large landslides, are predominantly developed nearly along these faults and influenced by river erosion (Figures 7 and 13). A typical large bedrock landslide such as the Atta Abad rock avalanche is commonly controlled by discontinuities related to the thrust fault activities, and the head-scarp regression and rainfall infiltration affect the stability [8].

In addition to the landslides, rock glaciers are abundant and typical nature hazards in the Karakoram region of northern Pakistan [74,75], although they have not been thoroughly studied [76]. The distribution of the rock glaciers in the entire Hunza basin compiled by visual analysis of high-resolution images is closely linked to the 0 °C isotherm between 3400 and 4600 m [36]. This zone is a universally high permafrost area in which the destabilized rock glaciers developed [74,75,77]; it is closely related to the climate changes and the increase in melt water attributed to the increase in temperature [78]. These characteristics

also explain the fundamental reason for the landslide developing on the slopes in a certain elevation range that is controlled mainly by the structures and interacts with the river and its erosion in the Hunza Valley.

The present study did not focus much on the rock glacier in the Hunza Valley, given that its deformation is related to the permafrost creep, especially in the climate changing area [79], and is not normally long-term and slow-moving, and is difficult to capture in the steep cliff covered by ice and permafrost. The potential risk from the rock glacier in the Hunza Valley should be further analyzed based on the previous studies [36,75]. The sediments of most rock glaciers in the glacier valley are transferred on a long journey of several to more than 10 km by the periodic flood along the channel of the glacier [8,11,36]. They pose potential threats to the river channels and infrastructures such as bridges [75,80].

## 5. Discussion

It is always challenging to identify slow-moving, active landslides, but these are often the largest and most catastrophic mass wasting events and material movements in mountainous areas such as the Hunza watershed [8]. Thus, early identification and prevention of landslides are equally significant missions compared with hazard mitigation.

In the Hunza Valley, even in northern Pakistan, multiple remote sensing techniques including optical remote sensing and InSAR have been utilized in landslide mapping and detection for hazard assessment [6,8,25,81]. However, there are defects in the present progress because the data used are relatively unitary, and the landslide dynamics development analyses are mostly based on historical landslides and, especially in the Hunza Valley, they have not been comprehensively discussed. This can be attributed to the lack of comprehensive landslide inventory data [35].

Compared with these studies, on the one hand, the present study took advantage of the multi-azimuth perspective of the ascending and descending Sentinel-1 datasets, thus monitoring the more comprehensive ground surface deformation in the Hunza Valley. The supplements of the literature and image interpretation were used indispensably. Consequently, the high-deformation-rate regions were captured by SBAS-InSAR in most areas in the Hunza Valley. On the other hand, this comprehensive inventory reports the latest landslides; thus, it ensures the up-to-date reality and utilization values of the landslide database for the Hunza Valley. The landslide identification based on the SBAS-InSAR monitoring and field investigation aimed to update the landslide inventory that had the potential of failure and the risk of leading to loess in the future. In this designed study, we achieved the goal of conducting the deformation monitoring and reconnaissance survey of landslides in the Hunza Valley. It is the advantage of this study that it can provide the datasets for related in-depth global research in the future.

As analyzed in the results, the landslides detected in this study were fundamentally validated by the field survey and the previous landslide reports [25,81]. This study also found that the landslide inventory in Hunza Valley was much more complete in terms of quantity and quality compared with the previous study conducted by Rehman et al. and Hussain et al. [25,36]. SBAS-InSAR results indicated deformation velocity and determined the active landslides, which contributed to the landslide assessment and exploration of the landslide distortion mechanism. By fitting the landslide area and volume data reported, the rule of power law exponent was constructed for the magnitude predication in this region. This study achieved landslide deformation pattern classification and concluded that the InSAR-detected landslides experienced significant deformation and had the potential for further acceleration and failure. The comprehensive inventory was intended to improve the support for in situ landslide monitoring and hazards management. Analysis of the landslide development characteristics and deformation patterns in different basin scales has improved our understanding of landslide hazards and called attention to the risk analysis of the population-densified Hunza Valley.

We cannot ignore that the frequent changes and colluvial infilling could mollify landslide boundaries [8] and admit that there are potential landslides that will become

active in the coming stage and moderate the difference between individual boundaries and the actual condition. Meanwhile, the findings demonstrated that, in this study, deformation monitoring in the mountainous alpine area may be affected by the steep terrain and dense vegetation cover. Specifically, owing to topography conditions such as extremely steep slopes and the environmental limitations of the glacier, ice, and dense vegetation, the Sentinel-1 datasets had lower coherence over the higher alpine area, resulting in the limited availability of coherence targets in these areas. In the future, multiple band SAR datasets with high resolution and high retravels should be applied. It will be meaningful to reconstruct the deformation and sliding process and to explore the mechanism related to landslide failure and the landslide–dam–flood hazards chain while taking landslides such as the Atta Abad landslide in 2010 as the typical case.

## 6. Conclusions

The Hunza Valley northwest of the Karakoram Mountains in northern Pakistan, which is a densely populated area with many villages and towns, is vulnerable to landslide disasters. This study successfully applied SBAS-InSAR and multi-track ascending and descending Sentinel-1A SAR datasets in the Hunza Valley to monitor the earth surface deformation velocity and completed the comprehensive, updated landslide inventory and development characteristics analysis combined with a field survey, image interpretation, and literature review. The maximum ground surface displacement velocities along the slope were calculated as $-311$ mm/y and $-490$ mm/y, based on the deformation velocity along the line of satellite sight ($V_{los}$) derived by SBAS-InSAR, respectively, from ascending and descending datasets. The slope that was acknowledged as consistent with the monitoring results was inventoried in the landslide field investigation. The comprehensive, updated inventory of 118 landslides, including the 53 latest detected active landslides and 65 landslides cited in the literature review and field survey, was completed. These landslides were numbered and named in the Hunza Valley, and the rule of power law exponent of landslide and volume were fitted using reported data. This study revealed that these landslides were predominantly characterized by a chair-like back wall, cracks, fragmented rocks, and deposits; additionally, the absence or near-absence of vegetation cover on the talus deposits was also evidence of active rock falls. The trends of the landslide deformation process were classified into constant rate, accelerating, decelerating, and fluctuating. Analysis of spatial development revealed that the aspects of the south, southwest, and west were the dominant sections of the landslide development. The landslides interpreted by the optical images from the literature (LL) had the unique features of lower velocity, area, elevation, and height. The landslides detected by Sentinel (LA, LD, and LB) had the special attributions of higher area, predicted volume, velocity, elevation, height, slope, and NDVI. The compiled landslides in this study mainly occurred in the areas with elevation between 1997 and 3379 m, under the zero-degree isotherm of 3400 m in the Hunza Valley. Except for the fault activities, the seasonal freeze–thaw procession in this altitude interval and the snow and ice melting above the higher elevation should be further analyzed since they are two factors connected to landslide failure, especially large-scale landslides. Large landslides have development features of high altitude, high position, and concealment, and the fault impact as controller and the inducing effect of river erosion are the two main causing factors. An updated recognition of landslide identification and its development characteristics will be useful for promoting scientific mitigation not only in the local communities but also in the China–Pakistan Economic Corridor and the larger Himalaya–Karakoram–Hindukush (HKH) region.

**Author Contributions:** Conceptualization, X.S. and Y.Z.; methodology, X.S.; software, X.S.; validation, X.S., X.M., M.U.R. and Z.K.; formal analysis, X.S.; investigation, X.S.; X.M., M.U.R., Z.K. and D.Y.; resources, X.S.; data curation, X.S. and M.U.R.; writing—original draft preparation, X.S.; writing—review and editing, X.S., Y.Z, X.M., M.U.R. and Z.K.; visualization, X.S.; supervision, Y.Z. and X.M.; project administration, X.M. and Y.Z.; funding acquisition, X.M. and Y.Z. All authors have read and agreed to the published version of the manuscript.

**Funding:** This study was supported by the Second Tibetan Plateau Scientific Expedition and Research (grant No. SQ2021QZKK0201), the important talent project of Gansu Province (grant No. 2022RCXM033), the National Natural Science Foundation of China (grant Nos. 41661144046, 42007232), the Science and Technology Planning Project of Gansu Province (grant No. 18YF1WA114), the Science and Technology Major Project of Gansu Province (grant No. 19ZD2FA002), and the Fundamental Research Funds for the Central Universities (grant No. lzujbky-2021-ey05).

**Informed Consent Statement:** Informed consent was obtained from all subjects involved in the study.

**Acknowledgments:** The Sentinel-1A images were provided by the European Space Agency (ESA) (https://scihub.copernicus.eu/, accessed on 20 July 2020). The National Aeronautics and Space Administration (NASA) provided the Landsat 8 images (accessed on 26 August 2020).

**Conflicts of Interest:** The authors declare that they have no known competing financial interests or personal relationships that could have appeared to influence the work reported in this paper.

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
