# Peer review of "Updating Inventory, Deformation, and Development Characteristics of Landslides in Hunza Valley, NW Karakoram, Pakistan by SBAS-InSAR"

_remotesensing, doi:10.3390/rs14194907_

Round 1

Reviewer 1 Report

This manuscript is based on SBAS-InSAR to identufy landslide inventory considering the defor- mation velocity and pattern of deformation process. It is of great significance to landslide disaster early warning. The following suggestions for revision are proposed for some of the research content of this paper.

(1) Line 65 “On 4 January 2010, a huge landslide caused 20 fatalities happened at Atta Abad and formed a 120 m high dam [6,7]”.What are the main triggers of this landslide? Gravity, earthquake or rainfall.

(2) Line 128 Landslide, in general, is one kind of geology process that experiences slow-moving deformation and eventually slide along the slip surface such as a soft layer or joint plane.The above description is only for gravity-triggered landslides. This is not the case for landslides triggered by factors such as earthquakes or rainfall. It is recommended to modify the relevant description.

(3) Line 333 Firstly, an annual deformation velocity of 20 mm/yr was set as threshold for the detection and mapping of potential active 335 landslides.What is the basis for setting a deformation velocity of 20 mm/yr as the threshold for potential active landslides.

(4) Line 383 The deformation velocity along the line of sight (Vlos) from ascending and descending Sentinel-1 datasets was detected using SBAS-InSAR, setting 0.3 and 0.4 as the coherence threshold respectively.What is the basis for setting 0.3 and 0.4 as the coherence threshold respectively.

(5) Line 884 The compiled landslides in this study are mainly occurring in the areas with elevation between 1997 and 3379 m, under the zero-degree isotherm of 3400 m in Hunza Valley.What are the main factor for large-scale landslides to be further analyzed under this condition?

(6) There are still some things to be improved in this manuscript:In Figure 2, line 243, the symbol after b is different from others.The font of 3.1 in line 261 is different from that of 3.2 in line 310.The serial number of this article is confusing. The serial number of line 253 is 1. The sequence number of line 379 is still 1. Line 803 and Line 857 are the same. Try to rearrange the sequence number.Line 524 (1), line 580 (2), and line 626 (3) have different formats and are not aligned.A large number of words are missing on the right side of page 22.There are a and b in the description of line 621, but there are no a and b in Figure 11.Line 748, the use of symbols after line 8C is inconsistent with the previous one.It is suggested to add a brief introduction to the acquisition of each data in the landslide database.The author can add a brief description of the advantages of SBAS-InSAR compared with others.When the abbreviation NDVI appears for the first time, there is a lack of explanation of its actual meaning.

(7) It is suggested to add an innovative description of this method.The conclusion can be more concise, the points can be more organized.

Author Response

Dear Editor and Reviewer,
We are very grateful for the comments and suggestions made by the referees. We have now made a first revision of the manuscript thoroughly according to the reviewer’s comments and suggestions. All the revised descriptions are marked, and highlighted in yellow . Some figures and references are also modified accordingly in the revised manuscript. The point-by-point responses are listed below, and associated revisions have been incorporated into the revised manuscript. And the manuscript was polished by native English speakers.

Yours sincerely,

Yi Zhang

General comments: This manuscript is based on SBAS-InSAR to identify landslide inventory considering the deformation velocity and pattern of the deformation process. It is of great significance to landslide disaster early warning. The following suggestions for revision are proposed for some of the research content of this paper.

(1) Line 65 “On 4 January 2010, a huge landslide caused 20 fatalities happened at Atta Abad and formed a 120 m high dam [6,7]”. What are the main triggers of this landslide? Gravity, earthquake or rainfall.

Response: Thanks for your question, yes, it is important to point out the triggers of this landslide. Based on the relative study about this landslide, the triggers of this landslide are thrust fault activities. On the one hand, based on the literature review, we learned that “The landslide was triggered by the movement of an active fault” (Ahmed, et al. 2016). On the other hand, as we analyse in 4.3, a typical large bedrock landslide, like the Attabad rock avalanche is commonly controlled by discontinuities related to the thrust fault activities, and the head-scarp regression and rainfall infiltration have an effect on the stability (Ahmed, et al. 2016). Thus the triggers of this landslide are thrust fault activities. And we disciplined that clearly in page 2 lines 15-17 and page 26 lines 24-28.

Bacha, A.S.; Shafique, M.; van der Werff, H. Landslide inventory and susceptibility modelling using geospatial tools, in Hunza-Nagar valley, northern Pakistan. Journal of Mountain Science 2018, 15, 1354-1370, doi:10.1007/s11629-017-4697-0.

Cook, N.; Butz, D. The Atta Abad Landslide and Everyday Mobility in Gojal, Northern Pakistan. Mountain Research and Development 2013, 33, 372-380, doi:10.1659/mrd-journal-d-13-00013.1.

Ahmed, M.F.; Rogers, J.D.; Bakar, M.Z.A. Hunza river watershed landslide and related features inventory mapping. Environmental Earth Sciences 2016, 75, 523, doi:10.1007/s12665-015-5172-2.

(2) Line 128 “Landslide, in general, is one kind of geology process that experiences slow-moving deformation and eventually slide along the slip surface such as a soft layer or joint plane.” The above description is only for gravity-triggered landslides. This is not the case for landslides triggered by factors such as earthquakes or rainfall. It is recommended to modify the relevant description.

Response: Thanks for your comments. Yes, you are right, the description of the landslide was modified based on the literature review in page 3 lines 30-36. A landslide, in general, is a geological process that causes a slow-moving deformation over a long period or a rapid, large deformation in a relatively short time. It eventually experiences failure in the types of slide, flow, fall, and topple along the slip surface or structures such as a soft layer or joint plane under var-ious causes such as gravity, earthquakes, rainfall, and human activity or a combination of these factors [27] (Hungr et al.,2014).

Hungr, O.; Leroueil, S.; Picarelli, L. The Varnes classification of landslide types, an update. Landslides 2014, 11, 167-194, doi:10.1007/s10346-013-0436-y.

(3) Line 333 “Firstly, an annual deformation velocity of 20 mm/yr was set as a threshold for the detection and mapping of potential active landslides.” What is the basis for setting a deformation velocity of 20 mm/yr as the threshold for potential active landslides?

Response: Thanks for your constructive comments. In this study, the detected deformation velocity is the basis for providing the target region (we can understand it as slopes) for further field survey and identifying the active landslides step by step.

Firstly, we check the detected deformation velocity with the reference of the geomorphology features from optical images (e.g. the Google Earth platform), and we have the assumption that the landslide will not occur on flat ground and on a gentle slope. Secondly, the preliminary mapping of suspected active landslides was carried out by superimposition on optical remote sensing images with reference to surface geomorphological features (e.g., scarps, sliding masses, and bulging toes). The inquiry and exploration of the deformation velocity threshold can be conducted in the ArcGIS platform on the basis of the spatial distribution of deformation velocity and the geomorphology features in images and DEM.

This way, we modified the illustration in page 9 in lines 38-45 in this revision: “Firstly, an annual deformation velocity of -20 mm/y along the slope direction was set as the threshold for distinguishing the relative stable area and the suspected active slope in which the landslide will occur. Secondly, the preliminary mapping of suspected active landslides was conducted by superimposition on optical remote sensing images with reference to surface geomorphological features (e.g., scarps, sliding masses, and bulging toes). Finally, …”

Some part of these steps also has been proven by several published studies (Liu et al. 2018, 2021), and we have cited these papers in this revised manuscript.

Liu, X.; Zhao, C.; Zhang, Q.; Peng, J.; Zhu, W.; Lu, Z. Multi-Temporal Loess Landslide Inventory Mapping with C-, X- and L-Band SAR Datasets-A Case Study of Heifangtai Loess Landslides, China. Remote Sensing 2018, 10, doi:10.3390/rs10111756.

Liu, X.; Zhao, C.; Zhang, Q.; Lu, Z.; Li, Z.; Yang, C.; Zhu, W.; Liu-Zeng, J.; Chen, L.; Liu, C. Integration of Sentinel-1 and ALOS/PALSAR-2 SAR datasets for mapping active landslides along the Jinsha River corridor, China. Engineering Geology 2021, 284, doi:10.1016/j.enggeo.2021.106033.

(4) Line 383 “The deformation velocity along the line of sight (Vlos) from ascending and descending Sentinel-1 datasets was detected using SBAS-InSAR, setting 0.3 and 0.4 as the coherence threshold respectively.” What is the basis for setting 0.3 and 0.4 as the coherence threshold respectively.

Response: Thanks for your comments. Although we applied less than 50 images in less than two years in temporal. Due to the special condition in our study area, the extra higher terrain relief, and various landforms and land cover, there are limitations in the acquire interferograms with high coherence in spatial scale. As shown in the processing products and results of InSAR monitoring, the decorrelation in the higher region that is covered by snow, ice, and glacier limits the detection of ground surface deformation and brings more noise and uncertainty. It is important to choose a relatively reliable threshold of coherence to derive enough deformation information in spatial and lead to not much noise in the results. Similarly, a similar threshold of 0.4 was set in the estimation of velocity in the ground subsidence monitoring of the loess plateau in the Yan’an New District by Hu, et al (2021). We modified it in page 10 in lines 41-46.

Hu, X., Xue, L., Yu, Y., Guo, S., Cui, Y., Li, Y., & Qi, S. (2021). Remote sensing characterization of Mountain Excavation and City Construction in Loess Plateau. Geophysical Research Letters, 48, e2021GL095230.  https://doi.org/10.1029/2021GL095230

(5) Line 884 “The compiled landslides in this study are mainly occurring in the areas with an elevation between 1997 and 3379 m, under the zero-degree isotherm of 3400 m in Hunza Valley.”What are the main factor for large-scale landslides to be further analyzed under this condition?

Response: Many thanks for your questions and inspiring us. As we discussed in page 26 in lines 1-7 in the revised manuscript. As to the reason for the clustering of landslides in this elevation interval of 1997 to 3379 m, except for the near proximity of river erosion and fault activities, the terrain in this section has less snow cover, is infiltrated by high-altitude meltwater, and experiences the process of seasonal freeze-thaw, which means these slopes are more likely to experience weathering and failure. Please read in page 26 in lines 1-7 in the revised manuscript.

In the nest future research, except for the fault’s activities, the seasonal freeze-thaw procession in this altitude interval and the snow and ice melting above higher elevations should be further analyzed since they are two factors that have connections with landslide failure, especially large-scale landslides. Please read the modifications of this point in page 29 lines 11-15 in the revised manuscript.

(6) There are still some things to be improved in this manuscript: In Figure 2, line 243, the symbol after b is different from others. The font of 3.1 in line 261 is different from that of 3.2 in line 310.The serial number of this article is confusing. The serial number of line 253 is 1. The sequence number of line 379 is still 1. Line 803 and Line 857 are the same. Try to rearrange the sequence number. Line 524 (1), line 580 (2), and line 626

Response: Thanks for your suggestions. It is our mistake. We uploaded the manuscript without using the template. We have checked it carefully and modified all the figures and sequence number correctly.

(7) have different formats and are not aligned. A large number of words are missing on the right side of page 22. There are a and b in the description of line 621, but there are no a and b in Figure 11. In line 748, the use of symbols after line 8C is inconsistent with the previous one. It is suggested to add a brief introduction to the acquisition of each data in the landslide database. The author can add a brief description of the advantages of SBAS-InSAR compared with others. When the abbreviation NDVI appears for the first time, there is a lack of explanation of its actual meaning.

Response: Thanks for your great comments and suggestions. It is our mistake due to upload the manuscript without using the template.

We have checked and modified correctly the formats and title of Figure 11.

As your suggestions, we add a brief introduction to the acquisition of each data in the landslide database, and a brief description of the advantages of SBAS-InSAR compared with others. In page 7 lines, 8-18: The time-series SBAS-InSAR technique with its deformation-detection ability that was proposed and developed by Berardino et al. and Lanari et al. [42-44] was applied and demonstrated in studying surface deformation and analysis in various fields, including ground subsidence detection, landslide identification, and glacier tracking [26,45-47]. Compared with PS-InSAR (Persistent Scatters Interferometry), SBAS-InSAR has a greater capability of monitoring deformations over the rugged mountainous area [48] and can provide valuable information for detecting surface deformation and time-series characteristics for analyzing the pattern and cause of active landslides [49,50].

We have checked all the pages of the manuscripts and added the full statement of the first appearing abbreviations including KKH, NDVI, and so on.

(8) It is suggested to add an innovative description of this method. The conclusion can be more concise, the points can be more organized.

Response: Thanks for your good suggestions. Combined with the whole modification of our manuscript based on reviewers’ suggestions, we have stated overall the innovative ingredients including the method, ideology, and conclusion in this revised paper. And the conclusion has been reorganized and drawn concisely.

Author Response

We are very grateful for the comments and suggestions made by the referees. We have now made a first revision of the manuscript thoroughly according to the reviewer’s comments and suggestions. All the revised descriptions are marked and highlighted in the revised manuscript. Some figures and references are also modified accordingly in the revised manuscript. The point-by-point responses are listed below, and associated revisions have been incorporated into the revised manuscript.

Yours sincerely,

Yi Zhang

(1) The abbreviation appearing for the first time in the text should state the full name, such as KKH on line 144.

Response: Thanks for your comments. We have checked all the pages of the manuscripts and added the full statement of the first appearing abbreviations including KKH, NDVI, and so on.

(2) It is suggested to put the corresponding figure at the back of the corresponding paragraph,

such as Figure 3 in front of 3.1 and Table 1 at the back of the second paragraph in 3.1. At present, typesetting reading is difficult.

Response: Thanks for your great suggestions. We have modified the arrangement of figures and tables.

(3) Figure 3 can be divided into two columns. The height of the current Figure 3 is too high. On the one hand, it occupies the layout, on the other hand, it affects the beauty.

Response: Thanks for your great suggestions. We have modified Figure 3 into two columns for convenient reading.

(4) In lines 288 and line 289, why is the spatial baseline of the data of ascending orbit set to 210, and the spatial baseline of descending orbit set to 200? What is the basis?

Response: Thanks for your great comments. In the SBAS-InSAR processing, the interferograms with small temporal and spatial separation (baseline) between the orbits were generated and applied in order to limit the noise effects referred to as decorrelation phenomena (Ferretti et al., 2001, Lauknes et al., 2011).

Ferretti, A., Prati, C., Rocca, F., 2001. Permanent scatterers in SAR interferometry. IEEE Trans. Geosci. Remote Sens. 39, 8–20. https://doi.org/10.1109/36.868878.

Lauknes, T.R., Zebker, H.A., Larsen, Y., 2011. InSAR deformation time series using an L1-norm small-baseline approach. IEEE Trans. Geosci. Remote Sens. 49 (1), 536–546. https://doi.org/10.1109/tgrs.2010.2051951.

(5) Are the incident angles of the lifting rail data in Table 1 all 37.05? Please verify.

Response: Thanks for your great comments and suggestions. The author checked the data used in this study and gave the incidences of LOS for both paths data and the average value of that in table 1.

(6) In line 316, "in-situ surveys" is changed to "Field investigations", and other places are similar.

Response: Thanks for your great suggestion. We have modified the statement in revised manuscripts.

(7) Line 333 sets the deformation rate to 20. What is the basis? The landslide will also occur because the deformation rate in some areas is less than 20, and the deformation rate in this area should be absolute.

Response: Thanks for your comments. You are right, that the landslide will also occur in an area the deformation rate is less than 20 mm/yr. On the one hand, in this study, we acknowledge that the SBAS-InSAR cannot detect all the landslides by monitoring ground-surface deformation. And some slopes will be sliding although their deformation velocity is less than 20 mm/yr. Thus, we combine the SBAS-InSAR monitoring, optical image interpretation, and literature review together to identify active landslides and supply other landslides with features. In this way, we accomplish one of the goals of this study: updating landslide inventory. On the other hand, it authors mistake misstatement of the first step about the use of deformation velocity. Based on the design process of our study, this work used displacement information to identify and delineate the landslide step by step. Some part of these steps has been proven useful for detecting and identifying active landslides [51,52].

Firstly, an annual deformation velocity of -20 mm/y along the slope direction was set as the threshold for distinguishing the relative stable area and the suspected active slope in which the landslide will occur. Secondly, the preliminary mapping of sus-pected active landslides was conducted by superimposition on optical remote sensing images with reference to surface geomor-phological features (e.g., scarps, sliding masses, and bulging toes). Finally, a field investigation was conducted to delineate the landslide mapping. In the field survey, landslide characteristics including topographic features, deformation evidence such as cracks, fissure, scarp, and the depositions were the criteria to ver-ify the SBAS-InSAR monitored displacement information; the results will be presented in Section 4.1.

We have modified that in the revised manuscripts. Please read page 9 in lines of 38-50. Hope the responses is clear for your comments

(8) Please check the sign of -20 mm/yr in line 336. The deformation rate of landslide position is not necessarily negative.

Response: Dear reviewer, yes, you are right. In some cases, the velocity along the line of sight could not directly indicate the actual movement of slope materials. In the case of landslide detection, most landslide or ground surface deformation occurs along the direction of the slopes, and therefore the deformation velocity along the slope (Vslope) was calculated for both ascending and descending paths with reference to the methods in the literature [17,51,52]. In the converted deformation velocity along the slope, the deformation velocity is negative. Please read page 9 in lines 6-10 in the revised manuscript.

Firstly, an annual deformation velocity of -20 mm/y along the slope direction was set as the threshold for distinguishing the relative stable area and the suspected active slope in which the landslide will occur. Secondly, the preliminary mapping of suspected active landslides was conducted by superimposition on optical remote sensing images with reference to surface geomorphological features (e.g., scarps, sliding masses, and bulging toes).

As explained in the response of comment 7 above. This misstatement was modified, as in page 9 lines 38-45.

(9) In line 385, "Vlos" is changed to "Vlos"

Response: Thanks for your comment and suggestion. This miswriting was modified.

(10) Line 385 sets the coherence threshold to 0.3 and 0.4, respectively. Why do you set the same threshold?

Response: Thanks for your comments. In this present study, although we applied less than 50 images in less than two years in temporal. Due to the special condition in our study area, the extra higher terrain relief, and various landforms and land cover, there are limitations in the acquire interferograms with high coherence in spatial scale. As shown in the processing products and results of InSAR monitoring, the decorrelation in the higher region that is covered by snow, ice, and glacier limits the detection of ground surface deformation and brings more noise and uncertainty. It is important to choose a relatively reliable threshold of coherence to derive enough deformation information in spatial and lead to not much noise in the results. And comparatively, for the ascending and descending Sentinel data, the coherence for the former is relatively lower than the later, thus we set the coherence threshold a little higher for descending data to generate more precise deformation results. With the reference, we learned that a similar threshold of 0.4 was set in the estimation of velocity in the ground subsidence monitoring of the loess plateau in the Yan’an New District by Hu, et al (2021).

Hu, X., Xue, L., Yu, Y., Guo, S., Cui, Y., Li, Y., & Qi, S. (2021). Remote sensing characterization of Mountain Excavation and City Construction in Loess Plateau. Geophysical Research Letters, 48, e2021GL095230.  https://doi.org/10.1029/2021GL095230

(11) The description of "It can …moraine" in lines 392-398 is confusing, the positive sign is inaccurate, and the meaning expressed by the author is unclear. Please modify it.

Response: Thanks for your comments. We have modified that explanation in page 11 lines 1-2.

(12) Does "439.92 km-2 and 557.06 km-2 " in line 402 indicate density? Why is there a decimal point for the number of dot densities? And the writing of "km-2 " is not proper.

Response: Thanks for your comments. Yes, the numbers indicate density. As per your suggestions, we modified the data and statement as in page 11 lines 6-7 in the revised paper. The densities of coherent points were calculated as 439 points and 557 points per square kilometer in the study area with an area of 1252.6 km2 for the ascending and descending datasets.

(13) What does "a.s.l" mean in line 405?

Response: Thanks for your comments. It just means “above sea level”, but it is a misleading expression, so we modified it.

(14) The color bands in Figure 5 and Figure 6 are too rough by inequality. Please use hierarchical representation.

Response: Thanks for your comment and suggestion. We have modified figure 5 and 6.

(15) What does "high-velocity values" mean in line 448? Please modify this statement.

Response: Thanks for your comments. We modified the statement as page 13 in lines 10-14 in the revised manuscripts. The interpretation of the velocity along the slope and optical images ensures the preliminary delineation of the boundary with the reference of deformation velocity with relatively high values, topographic characteristics (obtained from DEM), and the optical images’ interpreted features.

(16) Lines 462-465 describe that 53 potential landslides were identified by SBAS-InSAR technology, and another 65 landslides were described according to other references. Why did SBAS InSAR technology only identify 53 landslides, while the other 65 landslides were not identified? Its recognition rate is less than half.

Response: Thanks for your comments. We have clearly explained this concern about landslide data and the goal of updating landslide inventory based on various methods in the modified manuscript.

Firstly, the SBAS-InSAR have a great capability of monitoring deformations over rugged mountainous area and can offer valuable information for identifying and analyzing active landslide (in page 7 in lines 14-18). Therefore, the landslide identification using SBAS-InSAR is on the basis of the detected deformation. As for the deformation gained in the study area, the detected displacement information covers the main part of the valley below the elevation of 3,500 m (Figure 5). Because of the topography conditions such as extremely steep slopes and the environmental limitations of the glacier, ice, and dense vegetation, the Sentinel-1 datasets have lower sensitivity over the higher alpine area, resulting in the limited availability of coherence targets in these areas (page 11 in lines 10-15). For this concern, as suggested by another reviewer, it will be an effective strategy to avoid using winter images for detecting relatively whole monitoring results with coverage of upper areas in a such mountainous region. In the near future research about the ground deformation and landslide movement and mechanism, we can apply this strategy to gain more detailed results over high altitudes.

Secondly, in this study, we take the advantage of the SBAS-InSAR for identifying active landslide which is experiencing or has experienced deformation in the present period during monitoring. The landslides identified by the Sentinel-1A dataset (LA, LD, and LB) are characterized by higher displacement velocities compared to other landslides derived from the literature (LL) (Figure 9f). This was the reason that these smaller landslides were excluded in the SBAS-InSAR-based landslide detection and identification: because they were relatively stable with extra local deformation or extra slow-moving velocity and smaller area. ( page 17 in lines 38-50). This identified landslide can be an innovative ingredient of the updating landslide inventory and can be a key objective for analysis in this study and in the next study. But these landslides we interpreted by images with the reference of the literature, they may be experiencing extra local deformation or extra slow-moving deformation, or even rapid deformation, or more possible in the statue of dormant. And we also acknowledged that there also are some potential landslides that will become active in the coming stage (page 28 in lines 8-10). The deformation monitoring in the alpine mountainous area may be affected by the steep terrain and dense vegetation cover. Specifically, due to the topography conditions such as extremely steep slopes and the environmental limitations of the glacier, ice, and dense vegetation, the Sentinel-1 datasets have lower coherence over the supper higher alpine area, resulting in the limited availability of coherence targets in these areas. In the future, and multiple band SAR datasets with high resolution, high retravels should be applied (page 28 in lines 11-20).

(17) There is no legend in Figure 8. From the display results, some landslide InSAR results are not typical, as shown in Figures 8(e) and (h).

Response: Thanks for your comments. Yes. We can find that two landslides were not typical. The figure 8 comparatively shows typical examples of detected landslides and two landslides that were interpreted based on images and field investigation in the inventory. These landslides were mostly covered by SBSA-InSAR-detected coherent targets. The first four landslides were the active landslides identified based on the SBAS-InSAR monitoring (Figure 8(a-f)). Because of the steeper slope or the actual inactive state, some landslides may not have strictly corresponded to the criteria of higher deformation velocity for identifying landslides and experienced lower rates of deformation. These landslides were experiencing deformation and occurred previously (in history) or had developed after the ancient events (Figure 8g-h). Please read in lines 27-39 in page 14.

(18) How is the deformation curve in Figure 9 obtained? The location of the landslide should be clearly described. In addition, I think fig. 9(b) shows the overall deceleration and fig. 9(c) shows the overall acceleration. Please verify.

Response: Thanks for your comment and suggestion. According to the deformation results monitored by SBAS-INSAR, the landslide deformation time series curve is obtained at the position where the landslide deformation activity is obvious and the deformation velocity is maximum.

Thanks for your reminder, fig. 9(b) and fig. 9(c) were rearranged correctly in the revised manuscripts.

(19) What does "Magnitude" in line 580 mean?

Response: Thanks for your comments. The "Magnitude" means the area and volume of the landslide. In the revised manuscripts, we explained that in the title in line 9 in page 18.

(20) The "VL" in line 596 is inconsistent with the formula. Please number the formula.

Response: Thanks for your suggestions. We have checked and modified it.

(21) Line 598, "To" is changed to "to".

Response: Thanks for your suggestions. We have checked and modified it.

(22) What data is "reported data" in line 600th?

Response: The “reported data” refer to the published datasets, from which we gain information on the volume and area of the landslides. In the revised manuscripts, we modified that as “Published data” in page 19 in line 4 , and in lines 7-9.

Yi, X.; Shang, Y.; Shao, P.; Meng, H. A dataset of spatial distributions and attributes of typical rockfalls and landslides in the China-Pakistan Economic Corridor from 1970 to 2020. Science Data Bank 2021, 6, 5-14, doi:10.11922/sciencedb.j00001.00294.

(23) How do you get "-311 mm/y and -490 mm/y" in line 864? And the unit should be unified with the previous one.

Response: Thanks for your comments. In response to this, we first checked the paper and supplied the formulation of the method we used to calculate the deformation velocity along the slope direction (Vslope) based on the deformation velocity along the sight of satellite (Vlos) in page 9 in lines 4-25. And then, the maximum value of velocity along the slope was analyzed.

Reviewer 3 Report

Dear authors, thank you for the manuscript. It regards InSAR monitoring of a landslide area in Northern Pakistan.

The article needs extensive English language revision, especially the endings (-ed, -ing), to be better comprehensive. It is very difficult to read in this form.

If there is low coherence in the upper areas because of snow, the solution would be to avoid using winter images.

- figure 2: it is not evident if the numbers (27, 107) correspond to track number, or to the number of images

- figure 2: please mark the Hunza River Basin also in figure (a)

- figure 3 needs also language revision

- line 288: spatial baselines of LESS THAN 70 d and 210 m

- line 346: were presented -> will be presented

- line 385, coherence thresholds: 0.3 or 0.4 are very low values with regard to less than 50 images in each dataset. However, the results shown (fig 5) look reliably (spatial consistency)

- conversion to downslope velocities: do you set a threshold to the sensitivity? i.e. if the slope is almost perpendicular to LOS, do you still calculate the downslope velocity?

- do you compare results from one dataset with the other one?

- figure 9 description: (b) is decelerationg, (c) is accelerating

Author Response

We are very grateful for the comments and suggestions made by the referees. We have now made a first revision of the manuscript thoroughly according to the reviewer’s comments and suggestions. All the revised descriptions are marked and highlighted in the revised manuscript. Some figures and references are also modified accordingly in the revised manuscript. The point-by-point responses are listed below, and associated revisions have been incorporated into the revised manuscript. And the manuscript has been polished by native English speaker from MDPI.

Yours sincerely,

Yi Zhang

Dear authors, thank you for the manuscript. It regards InSAR monitoring of a landslide area in Northern Pakistan.

1. The article needs extensive English language revision, especially the endings (-ed, -ing), to be better comprehensive. It is very difficult to read in this form.

Response: Thanks for your comments and suggestions. We have checked all the pages and all sentences and extensively polished the words and statements of the manuscripts.

2. If there is low coherence in the upper areas because of snow, the solution would be to avoid using winter images.

Response: Thanks for your suggestions. It will be an effective strategy to avoid using winter images for detecting relatively whole monitoring results with coverage of upper areas in a such mountainous region. In the near future research about the ground deformation and landslide movement and mechanism, we can apply this strategy to gain more detailed results over high altitudes. The study by Xie Hu et al also applies this kind of similar strategy in the ground deformation monitoring in Loess Plateau by removing several de-correlated pairs during the monsoonal summertime from May to September (Hu et el., 2021).

3. figure 1: it is not evident if the numbers (27, 107) correspond to track number, or to the number of images

Response: Thanks for your comments. The author checked and modified the legend of Figure 1 and all the explanations of the Sentinel-1 data. We applied the Sentinel-1 datasets from Path 27 (Ascending) and Path 107 (Descending) with their footprints overlapped in Hunza Valley.

4. figure 1: Please mark the Hunza River Basin also in figure (a)

Response: Thanks for your suggestion. The author labeled the Hunza Basin in figure 1(a) in page 5 in line 5 in the revised manuscripts.

5. figure 2 needs also language revision

Response: Thanks for your comments. We have extensively polished the words and statements of the explanation of figure 2.

6. line 288: spatial baselines of LESS THAN 70 d and 210 m

Response: Thanks for your reminder, we modified the miswriting.

7. line 346: were presented -> will be presented

Response: Thanks for your comments. We have modified that in page 9 in line 50 in revised manuscript.

8. line 385, coherence thresholds: 0.3 or 0.4 are very low values with regard to less than 50 images in each dataset. However, the results shown (fig 5) look reliably (spatial consistency)

Response: Thanks for your comments. Although we applied less than 50 images in less than two years in temporal. Due to the special condition in our study area, the extra higher terrain relief, and various landforms and land cover, there are limitations in the acquire interferograms with high coherence in spatial scale. As shown in the processing products and results of InSAR monitoring, the decorrelation in the higher region that is covered by snow, ice, and glacier limits the detection of ground surface deformation and brings more noise and uncertainty. It is important to choose a relatively reliable threshold of coherence to derive enough deformation information in spatial and lead to not much noise in the results. With the reference, we learned that a similar threshold of 0.4 was set in the estimation of velocity in the ground subsidence monitoring of the loess plateau in the Yan’an New District by Hu, et al (2021).

Hu, X., Xue, L., Yu, Y., Guo, S., Cui, Y., Li, Y., & Qi, S. (2021). Remote sensing characterization of Mountain Excavation and City Construction in Loess Plateau. Geophysical Research Letters, 48, e2021GL095230.  https://doi.org/10.1029/2021GL095230

9. conversion to downslope velocities: do you set a threshold to the sensitivity? i.e. if the slope is almost perpendicular to LOS, do you still calculate the downslope velocity?

Response: Thanks for your comments. Yes, we set a limited intersection in the conversion process. In the deformation monitoring along the sight of the satellite. The trajectories and imaging geometry determine the full foreshortening that occurs due to the incidence is equal to the slope gradient when the slope is almost perpendicular to LOS. It can not be avoid, so all the Vslope is based on the Vlos. But we explained the process of the conversion in this revised manuscript: 

The SBAS method obtains the surface deformation velocity along the line of sight (VLOS). From the interpretation of the relationship between slope movement and VLOS, a positive (negative) value indicates movement towards (away from) the satellite. However, in some cases, the velocity along the line of sight could not directly indicate the actual movement of slope materials. To further investigate and map surface deformation and geohazards, the velocity along the direction of the slope (VLOS) was converted to the deformation rate along the slope (Vslope), using formulas developed by Kang et, al, and Schlögel et, al. Please read in page 9 in line 4-25 in the revised manuscript.

We can know that the value of parameter Index can be positive or negative. The Vslope will be calculated as infinity when the Index is near to zero. Based on the statistic and previous in this study area, we set the value in the section of (-0.3 ~ 0) as -0.3, and set the value in the section of (0 ~ 0.3) as 0.3.

Kang, Y.; Zhao, C.; Zhang, Q.; Lu, Z.; Li, B. Application of InSAR Techniques to an Analysis of the Guanling Landslide. Remote Sensing 2017, 9, 1046.

Schlögel, R.; Doubre C.; Malet J.-P.; Masson F., Landslide Deformation Monitoring with ALOS/PALSAR Imagery: A D-InSAR Geomorphological Interpretation Method. Geomorphology, 2015. 231: p. 314-330.https://doi.org/10.1016/j.geomorph.2014.11.031

Zhao Fumeng. Early Identification of Geological Hazards and Landslides Susceptibility Evaluation of Karakorum Highway (Domestic Section) [D].LanzhouUniversity,2020.DOI:10.27204/d.cnki.glzhu.2020.000254.

10. do you compare results from one dataset with the other one?

Response: Thanks for your comments. Yes, we conducted the comparative analysis between the landslides identified by ascending and descending Sentinel-1A and also the landslides interpreted by the reference of literature in the results 4.2.1. And we also, explained that the difference and advantage and innovation of this study compared with the published research in Hunza Valley.

In our present study, these landslides identified by the multi tracks Sentinel-1A dataset are characterized by higher displacement velocities compared to other landslides derived from the literature (LL) (Figure 9f). Please read in page 17 in lines 38-41 in the revised manuscript.

It also can be found that SBAS-InSAR detected landslides are larger in volume compared to the landslides interpreted by reference (Fig-ure 10b). Please read page 19 in lines 14-16 in the revised manuscript.

Comparatively, there are defects in the present progress that the data used are relatively unitary, and the landslides dynamics development analysis is mostly based on the historical landslides and, especially in the Hunza Valley, it has not been comprehensively discussed. This can be attributed to the lack of comprehensive landslide inventory data (Su et al., 2021). Compared with these studies, on the one hand, the present study took advantage of the multi-azimuth perspective of the ascending and descending Sentinel-1A datasets, thus monitoring the more comprehensive ground surface deformation in Hunza Valley. Please read in page 27 in lines 19-23 in the revised manuscript.

As analysed in the results, fundamentally, the landslides detected in this study were validated by the field survey and the previous landslide reports (Rehman et al., 2020; Hassan et al., 2021). This study also claims that the landslides inventory in Hunza Valley is much more complete in terms of quantity and quality compared with previous study conducted by Rehman et al. and Hussain et al (Hussain et al., 2021). Please read in page 27 in lines 37-42 in the revised manuscript.

Rehman, M.U.; Zhang, Y.; Meng, X.; Su, X.; Catani, F.; Rehman, G.; Yue, D.; Khalid, Z.; Ahmad, S.; Ahmad, I. Analysis of Landslide Movements Using Interferometric Synthetic Aperture Radar: A Case Study in Hunza-Nagar Valley, Pakistan. Remote Sensing 2020, 12, doi:10.3390/rs12122054.

Su, X.; Zhang, Y.; Meng, X.; Yue, D.; Ma, J.; Guo, F.; Zhou, Z.; Rehman, M.U.; Khalid, Z.; Chen, G.; et al. Landslide mapping and analysis along the China-Pakistan Karakoram Highway based on SBAS-InSAR detection in 2017. Journal of Mountain Science 2021, 18, 2540-2564, doi:10.1007/s11629-021-6686-6.

Hassan, J.; Chen, X.; Muhammad, S.; Bazai, N.A. Rock glacier inventory, permafrost probability distribution modeling and associated hazards in the Hunza River Basin, Western Karakoram, Pakistan. Sci Total Environ 2021, 782, 146833, doi:10.1016/j.scitotenv.2021.146833.

Hussain, M.A.; Chen, Z.; Wang, R.; Shoaib, M. PS-InSAR-Based validated landslide susceptibility mapping along Karakorum Highway, Pakistan. Remote Sensing 2021, 13, doi:10.3390/rs13204129.

11. figure 9 description: (b) is decelerating, (c) is accelerating

Response: Thanks for your reminder. In the revised manuscript, the order of figures has been modified correctly in page 18 in line 1, the figure 9.

Round 2
